



# Measurement report: Three years of size-resolved eddy-covariance particle number flux measurements in an urban environment

Agnes Straaten, Stephan Weber

Institute of Geoecology, Technische Universität Braunschweig, Braunschweig, 38106, Germany

*Correspondence to*: Agnes Straaten (agnes.straaten@tu-braunschweig.de)

**Abstract.** Size-resolved particle number fluxes in the size range 10 nm < particle diameter ($D_p$) < 200 nm were measured over a 3-year period (April 2017 - March 2020) using the eddy covariance technique at an urban site in Berlin, Germany. The observations indicated the site as a net source of particles with a median total particle number flux of $F_{TNC}$ = 0.86 x $10^8$ $m^{-2}$ $s^{-1}$. The turbulent surface-atmosphere exchange of particles was clearly dominated by ultrafine particles ($D_p$ < 

100 nm) with a share of 96 % of total particle number flux ($F_{UFP}$ = 0.83 x $10^8$ $m^{-2}$ $s^{-1}$).

Annual estimates of median $F_{TNC}$ and $F_{UFP}$ slightly decreased by -9.6 % (-8.9 % for $F_{UFP}$) from the first to the second observation year and a further -5.9 % (-6.1 % for $F_{UFP}$) from the second to the third year. The annual variation might be due to different reasons such as variation of flux footprints in the individual years, a slight reduction of traffic intensity in the third year or a progressive transition of the vehicle fleet towards a higher share of low-emission standards or electric drive. Size-

resolved measurements illustrated events of bidirectional fluxes, i.e. simultaneous emission and deposition fluxes within the size spectrum, which occurred more often in spring, late summer and autumn than in winter. Multi-year observations of size-resolved particle fluxes proved to be important for deeper understanding of particle exchange processes with the urban surface and the pronounced influence of traffic at this urban site.

## 1 Introduction

Airborne particles, especially ultrafine particles (UFP) with diameters $D_p$ < 100 nm, may enter the circulatory system and are associated to acute and chronic effects on human health (Oberdörster et al., 1995; Nemmar et al., 2002; HEI Review Panel on Ultrafine Particles, 2013; Schraufnagel, 2020). UFP are abundant in cities due to a number of different anthropogenic emission sources such as private and public traffic or commercial and industrial combustion (Tsang et al., 2008; Tie et al. 2009; Bäfver et al. 2011; Weber et al., 2013; Kumar et al., 2014; Gerling et al., 2020). In order to understand the spatio-temporal variation

of population exposure to ultrafine particles, information on the strength and location of sources and sinks, and the turbulent exchange between surface and atmosphere are essential (Buzorius et al., 2000; Longley et al., 2003; Mårtensson et al., 2006; Weber et al., 2013).

The turbulent surface-atmosphere-exchange of pollutants may be quantified using the micrometeorological eddy covariance (EC) method (e.g., Baldocchi 2003; Burba and Anderson, 2005; Aubinet et al., 2012). During the past decades EC was applied





over non-urban and urban terrain to quantify gaseous pollutant and particle fluxes (e.g. Buzorius et al., 1998; Suni et al., 2003; Damay et al., 2009; Ripamonti et al., 2013; Gioli et al., 2013; Deventer et al., 2018).

Urban particle number flux measurements using EC with varying lower cut-off diameters of the particle counting instruments were conducted in a couple of cities across Europe (cf. Table 1). These studies generally report the city to be a net source of particles, i.e. a positive average particle number flux, which by convention indicates an upward directed emission flux to the

atmosphere in contrast to a negative (downward directed) deposition flux. A significant contribution to the particle number flux is attributed to vehicle traffic with peak mode diameters of fresh traffic emission in the size range $D_p < 30$ nm (Wiedensohler et al., 2002; Zhu et al., 2007; Rönkkö et al., 2017). On the diurnal cycle, particle number fluxes are relatively small at night but increase in the early morning hours due to anthropogenic activity and enhanced turbulence, resulting in peak upward fluxes at around noon or in the early afternoon (Dorsey et al., 2002; Mårtensson et al., 2006; Deventer et al., 2013;

Conte et al., 2018). Variation in particle number fluxes is shown to be related to different land use within the flux source area with lower emission fluxes from vegetated surfaces or residential areas and higher fluxes from urban land-use with intense traffic (e.g. Järvi et al. 2009, Ripamonti et al. 2013).

Although aerosol processes and dynamics such as particle deposition or the influence on human health are strongly related to the particle diameter, information on size-resolved urban particle number fluxes, especially in the ultrafine size range, remains

scarce. Size-resolved fluxes, however, help to deepen understanding on particle sources and dynamics in the urban boundary layer. Additionally, the flux data is essential to parameterise dry deposition velocities and to validate urban air quality models (Saylor et al., 2019; Farmer et al., 2021). Size-resolved flux observations may point to situations of bidirectional particle fluxes, i.e. simultaneously occurring upward and downward fluxes in different ranges of the particle size spectrum (Schmidt and Klemm, 2008; Deventer et al., 2015). Usually upward fluxes tend to occur in the smaller size ranges and may point to emission

sources such as vehicle traffic whereas downward fluxes mainly occur in larger size ranges, i.e. accumulation mode particles. Particle flux studies usually comprise observation periods of a few weeks or months within different seasons of the year. Only few studies report observations for longer periods such as one-year or multi-annual periods (e.g. Ripamonti et al., 2013; Deventer et al., 2015, cf. Table 1). This is especially true for size-resolved particle number fluxes from urban areas for which multi-annual data has not been available, yet. These longer term studies, however, are important to analyse temporal variation

of urban particle number fluxes and their driving mechanisms.

Here, we report on three years of size-resolved urban particle number fluxes in the diameter size range 10 nm $< D_p <$ 200 nm. The measurements were carried out in central Berlin, Germany, in the framework of the urban climate research project [UC]²-3DO (Scherer et al., 2019). We analyse the diurnal, seasonal and annual variation of ultrafine and size-resolved particle number fluxes and study the contribution from surface drivers such as land-use on temporal variation of particle fluxes.




**Table 1: Overview of total particle number fluxes of measurement campaigns from ten different cities. Right column: The information in brackets specifies the data basis of the individual studies for estimating the observed particle fluxes.**

| City | Study | Duration/ Season | Size-resolved | Lower cut-off (nm) | Particle number fluxes ($10^8$ m$^{-2}$s$^{-1}$) |
|---|---|---|---|---|---|
| **Münster** | Schmidt & Klemm, 2008 | ~ 3 months, summer | yes | 30 | -0.43 – 1.44 |
| | Deventer et al., 2013 | 1-2 months, spring | yes | 55 | 0 – 0.1 (mean diurnal cycle) |
| | Deventer et al., 2015 | 1 year | yes | 55 | 0.02 – 0.1 (seasonal fluctuations) |
| **Innsbruck** | Deventer et al., 2018 | ~1 months, summer | yes | 6 | 0.5 – 3.0 [a] |
| | Heyden et al., 2018 | ~1 months, summer | no | 10 | -1.9 – 2.8 (10-90 percentile) |
| **Lecce** | Contini et al., 2012 | 1-2 month, spring | no | 9 | 2.0 – 10.8 [b] (IQR) |
| | Conte et al., 2018 | 1-2 months, spring | yes (3 size bins) | 8 | 2.82 [b] |
| **Helsinki** | Järvi et al., 2009 | 1 year | no | 6 | 0.44 – 8.4 [a] |
| | Ripamonti et al., 2013 | 3 years | no | 6 | 0.05 – 11 (daily variation) |
| | Kurppa et al., 2015 | 2 years | no | 6 | 0.1 – 8.2 (5-95 percentile) |
| **London** | Martin et al., 2009 | ~ 1 month, autumn | no | 10 | 0.5 – 6.5 (mean diurnal cycle) |
| | Harrison et al., 2012 | ~ 1 month, autumn | yes no | 50 10 | up to 3 0.5 - 7 |
| **Edinburgh** | Nemitz et al., 2000 | 3 weeks each, spring, autumn | no | 100 | 0.1 – 0.6 |
| | Dorsey et al., 2002 | 2 weeks each, spring, 2x autumn | no | 11 | 0.9 – 9 (mean diurnal cycle) |
| | Martin et al., 2009 | summer, autumn | no | 11 | 5 – 11 (mean diurnal cycle) |
| **Stockholm** | Mårtensson et al., 2006 | 2-3 months, spring | no | 11 | 0.1 - 3 |
| **Manchester** | Longley et al., 2004 | 2 weeks, autumn | yes (3 size bins) | 100 | 0.11 – 0.37 (mean diurnal cycle) |
| | Martin et al., 2009 | 1-1.5 months, summer, winter | no | 3 | 3 – 20 (mean diurnal cycle) |



| Gothenburg | Martin et al., 2009 | ~1 month, winter | no | 5 | 1 – 7 (mean diurnal cycle) |
| Berlin | this study | 3 years | yes | 10 | 0.2 – 2.0 (mean diurnal cycle) |

[a] average daytime emission, [b] measurement height 14 m (roughness sub-layer)

## 2 Materials and Methods

### 2.1 Measurement site

Size-resolved particle number fluxes were measured from 01 April 2017 to 31 March 2020 near "Ernst-Reuter-Platz" in central Berlin, Germany, atop the main building of Technische Universität Berlin. The site surroundings comprise residential built-up surfaces, traffic and green areas (Figure 1). To the north of the flux site, the busy main road "Straße des 17. Juni" with an average daily traffic intensity of about 37,300 vehicles day$^{-1}$ is located (Umweltatlas Berlin, 2017). To represent traffic intensity in the flux footprint, data from two traffic counting stations at "Hardenbergstraße" (HS, south-west of measurement site) and "Straße des 17. Juni" (S17J, east of the site) were available (cf. Figure 1). Data was provided by the traffic information centre Berlin (VMZ Berlin Betreibergesellschaft mbH) at hourly resolution.

### 2.2 Instrumentation

The particle flux instrumentation consisted of an electric mobility particle sizer (Engine Exhaust Particle Sizer Spectrometer, EEPS 3090, TSI Inc., Minnesota, USA) and a 3D ultrasonic anemometer (USA-1, Metek GmbH, Elmshorn, Germany). Both instruments synchronously sampled at a frequency of 10 Hz. The EEPS measures the particle number size distribution (PNSD) over the size range 5.6 nm $< D_p <$ 560 nm in 32 size channels. Through a stainless steel tube of 0.01 m diameter, the air was sampled at a flow rate of 10 L min$^{-1}$ resulting in a laminar sampling flow (Reynolds number ≈ 1300; Hinds, 1999). The steel tube was attached to a 10 m meteorological rooftop mast with the sample inlet located next to the sonic anemometer at a height of 57 m above ground level. We sampled dry aerosol using a Nafion dryer MD-700 (Perma Pure LLC, length 0.9 m) to keep relative humidity in sample air < 40 %.

The particle spectrometer EEPS 3090 classifies particles based on their differential electrical mobility. Charged particles entering the electric field are reflected outwards where the charge is delivered to 22 electrodes and converted into particle number concentrations (TSI Inc., 2015). The instrument was primarily developed for measurements of engine exhaust emissions that show higher particle concentrations than usually observed in the urban background. Hence, EEPS readings of specific size channels may fall below the analyser's minimum threshold concentration resulting in non-valid concentration readings. To avoid data gaps within the PNSD, a cubic natural spline interpolation was used as gap-filling strategy to obtain complete PNSDs (cf. section 2.3).





**Figure 1: Eddy covariance measurement site in central Berlin near Ernst-Reuter-Platz (ERP; data sources: Geoportal Berlin, 2014 (modified), 2021; Umweltatlas Berlin, 2017).**

To quality-check EEPS measurements for concentrations and fluxes, an on-site inter-comparison to a water-based condensational particle counter (WCPC 3787, TSI Inc., Minnesota, USA) was carried out.

## 2.3 Data handling

The post-processing of particle data and fluxes comprised the following procedures:

Size-dependent diffusional particle losses within the sampling line and the Nafion dryer were corrected according to Hinds (1999). The Nafion dryer MD-700 was assumed as stainless steel tube of 0.9 m length as no specific correction factors were available for the dryer.

Subsequently, gaps in the PNSD (cf. section 2.2) were filled using a natural spline interpolation following Meyer-Kornblum et al. (2019). As gap-filling caused larger uncertainties in the boundary regions of the PNSD, the size spectrum was limited to 21 size channels in the size range 10 nm < $D_p$ < 200 nm. According to Meyer-Kornblum et al. (2019), PNSDs were gap-filled as long as less than 9 gaps in total (minimum of 12 remaining data points within the PNSD) or 5 contiguous gaps (5





neighbouring size channels) occurred in that size range. PNSDs not fulfilling these requirements were discarded. The total
particle number concentration (TNC, 10 nm < $D_p$ < 200 nm), ultrafine particles (UFP, 10 nm < $D_p$ < 100 nm) and three modal
concentrations, i.e. nucleation mode (NUC, 10 nm < $D_p$ < 30 nm), Aitken mode (AIT, 30 nm < $D_p$ < 100 nm), and accumulation
mode (ACC, 100 nm < $D_p$ < 200 nm) were calculated from gap-filled PNSDs.

For particle flux calculation, the missing samples allowance for each half-hourly block average was set to 20 %. Wind vectors
and particle number concentrations were checked for plausibility concerning a realistic range of absolute values and spikes
were removed following Vickers and Mahrt (1997). Additionally, double coordinate rotation for tilt correction and spectral
corrections of high-pass (Moncrieff et al., 2004) and low-pass (Moncrieff et al., 1997) filtering effects were applied. For time
lag compensation between particle and sonic data, we used covariance maximization with a specified time lag window.
Following other urban particle flux studies, we applied linear detrending (e.g. Mårtensson et al., 2006; Vogt et al., 2011a;
Deventer et al., 2013; Deventer et al., 2018; Heyden et al., 2018). The particle flux calculation was performed using the
software EddyPro® v6.2.2. To calculate particle number fluxes for WCPC data (cf. section 2.2), the same procedure as
described above was adopted. By definition, positive flux values indicate upward (emission) fluxes to the atmosphere whereas
negative values denote downward (deposition) fluxes to the surface.

The response time of the EEPS for fast changes in concentration was reported as 0.5 s (Johnson et al., 2003, 2004). This
response time is similar to other CPCs or size spectrometers used in particle flux measurements (e.g. Buzorius, 2001; Dorsey
et al., 2002; Vogt et al., 2011b; Deventer et al., 2015). Limited sensor response is one reason for underestimation of turbulent
particle fluxes. We corrected sensor response according to Horst (1997) resulting in an average flux correction of between
+2.4 and +3.0 % in the 21 size channels. Finally, particle number fluxes with quality flags > 6 according to Foken et al. (2004)
were discarded from subsequent analysis. For the flux footprint prediction, we used the two-dimensional parameterisation of
Kljun et al. (2015).

**2.4 Data availability**

For the entire three-year period, data availability was 59.2 % for total particle number flux ($F_{TNC}$) (59.3 % for UFP flux, $F_{UFP}$).
For the individual observation years, data availability of $F_{TNC}$ decreased from 62.9 % ($F_{UFP}$: 63.0 %) in the first year to 61.1 %
($F_{UFP}$: 61.2 %) in the second and 53.6 % ($F_{UFP}$: 53.6 %) in the third year. The data availability varied due to different reasons
such as quality checks, PNSDs that not fulfil the requirements for gap-filling, and off-times due to maintenance of the EEPS.
The lower data availability in the third year was due to a 2-month period (22 May – 24 July 2019) during which the EEPS was
not available at the site due to inter-comparison measurements.

**2.5 Quantification of measurement uncertainty**

Atmospheric observations are prone to measurement uncertainty, e.g. sampling errors as well as sensor and gap-filling
uncertainty. In the following, we briefly outline the approach to quantify the potential sources of uncertainty.





The gap-filling of size channels below the minimum threshold concentration of the EEPS causes uncertainty due to the natural spline interpolation method (Meyer-Kornblum et al., 2019). Hence, percentage errors resulting from the interpolation process are dependent on the number of gaps within a PNSD and the particle diameter. The highest average percentage errors of 13.2 up to 31.6 % due to gap-filling occur in the size range of 10-20 nm. The corresponding uncertainty estimates for the size ranges of 20-50 nm, 50-100 nm, and 100-200 nm were between 5.6-17.2 %, 2.0-3.7 %, and 2.2-6.7 %, respectively (Meyer-Kornblum

et al., 2019).

  The random error for eddy covariance flux measurements was estimated according to Finkelstein and Sims (2001). The median random flux error varies between 27 and 40 % for the different size channels (25 % for $F_{UFP}$ and $F_{TNC}$) and generally increases towards the boundary size regions of the PNSD. This coincides with random flux errors of between 28 and 39 % as reported by Deventer et al. (2018).

**2.6 Land-use regression analysis**

  A multiple linear land use regression (LUR) analysis was carried out to estimate the relationship of $F_{TNC}$ and $F_{UFP}$ with land-use in the flux footprint. The land-use types were determined based on a biotope type mapping provided by city authorities (Geoportal Berlin, 2014). We binned data into 16 wind sectors, which consisted of three land-use types, namely "built-up areas", "traffic areas", and "green areas". The land-use types "water surfaces" and "other areas" were eliminated to avoid

collinearity due to high correlation (Pearson's $r \geq 0.6$ or $r \leq -0.6$) with other land-use types and the small amount of these surface types within the footprint. Since the model is thought to provide an estimate of the variation of particle emission and deposition with land-use, non-significant variables were not eliminated.



# 3 Results

## 3.1 Meteorological characterisation of the study period

For an overview of the meteorological situation in the measurement period, data from the German weather service station "Berlin Tempelhof" (7 km SE from the flux site) was used. The average annual air temperatures during the study period were 10.3 °C, 12.5 °C, and 12.0 °C (average annual temperature 1981-2010: 9.9 °C). In terms of annual precipitation, the first observation year (04/2017-03/2018) with an annual precipitation sum of 798 mm was significantly wetter than the long-term average of 577 mm (1981-2010). However, precipitation in the second year (04/2018-03/2019) and the third year (04/2019-03/2020) amounted to only 384 mm and 433 mm with the year 2018 characterised by a summer drought and hot spells. Wind at the "Tempelhof" site blows dominantly from SW (average wind direction in 1981-2010: 243°). The annual average wind direction only slightly varied with 255°, 272°, and 248° over the three years, respectively (Figure 2, Figure A1).

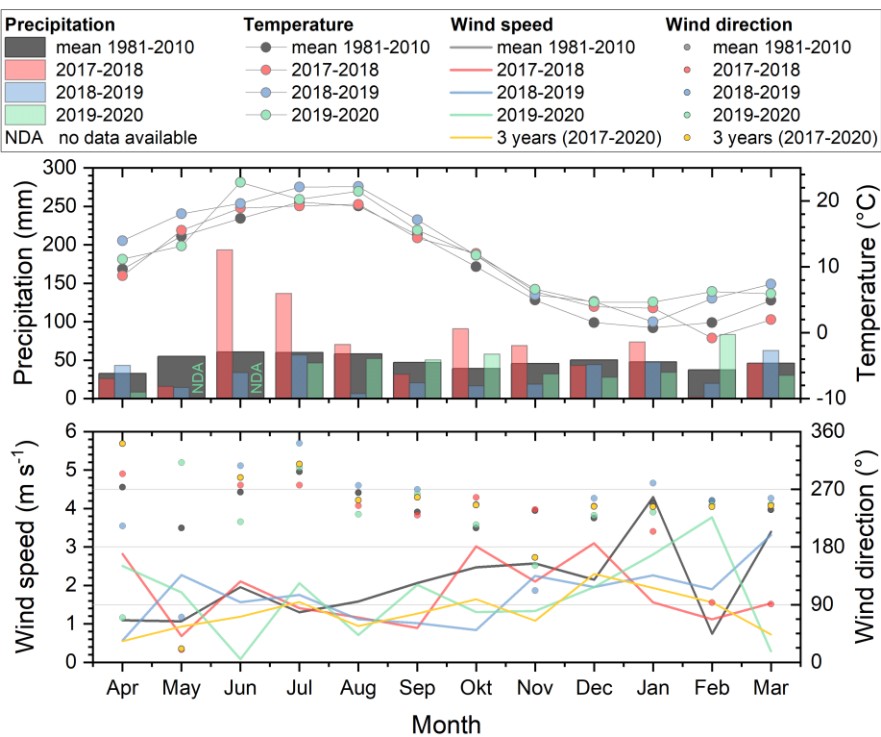

**Figure 2: Meteorological data such as temperature, precipitation, wind speed, and wind direction from the German weather service station "Berlin Tempelhof" (German weather service site ID: 433) located 7 km SE from the flux site.**





### 3.2 Data quality assurance and quality control

#### 3.2.1 Quality check of EEPS

The on-site inter-comparison of particle number concentration between EEPS and WCPC (cf. section 2.2) indicated good agreement with an average relative deviation of 10.6 % (slope 0.94, $R^2 = 0.84$, Figure A2a). It has to be noted that we compare the WCPC data (lower cut-off of 5 nm) with the gap-filled EEPS data (10 nm < $D_p$ < 200 nm). The deviation in particle concentrations is plausible as it is caused by different lower and upper cut-off diameters.

The time series of WCPC and EEPS fluxes were strongly correlated (r = 0.91), although the EEPS underestimated fluxes

measured by WCPC (slope 0.66, $R^2 = 0.84$, average relative deviation of 26 %, Figure A2b). We argue that the underestimation is due to the lower WCPC cut-off (5 nm) and the dominance of nucleation mode fluxes at the present site (cf. section 3.5).

#### 3.2.2 Spectral analysis

Spectral analysis is used in EC applications to study the frequency response of the measurement setup. The measured cospectra are either compared to ideal cospectra (e.g. Kaimal et al., 1972) or to the sonic sensible heat flux cospectrum which often

resembles an ideal cospectrum (Aubinet et al., 2012).

We calculated normalised daytime cospectra for $F_{UFP}$ for all wind directions under neutral stratification. Most of the cospectra values were positive (blue dots) indicating the frequent occurrence of emission fluxes (Figure 3). The particle flux cospectrum agreed well with sensible heat and the ideal Kaimal spectrum. Furthermore, the cospectrum follows the theoretical -4/3 slope in the inertial subrange indicating reasonable sensor frequency response.

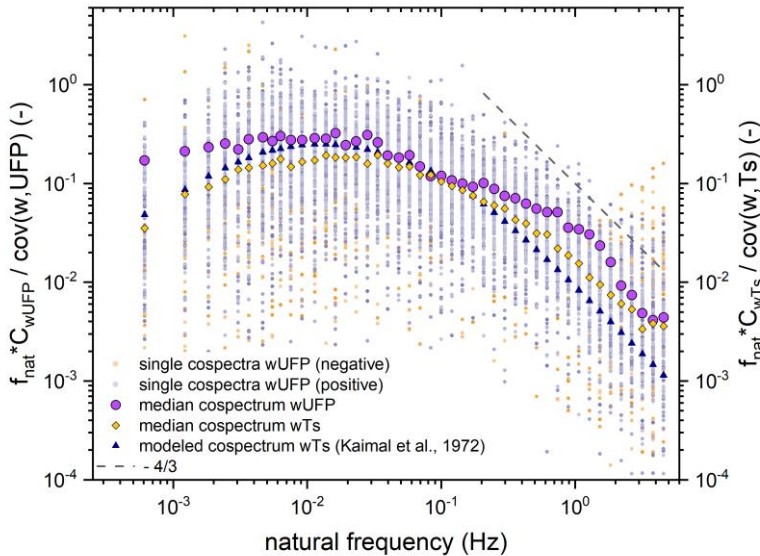


**Figure 3: Normalised (median) cospectra of $F_{UFP}$ (n = 160), sonic sensible heat flux, and ideal cospectrum after Kaimal et al. (1972). Additionally, the -4/3 slope in the inertial subrange is indicated. Blue dots represent positive values of the cospectrum, yellow dots negative values with inverse sign.**

### 3.3 Particle number concentrations

Over the course of the three-year study period, we observed median TNC concentrations of 7,300 cm$^{-3}$ and 6,450 cm$^{-3}$ for

UFP, respectively (cf. Table 2). The half-hourly average TNC varied between a minimum of 3,099 cm$^{-3}$ and a maximum of

53,879 cm$^{-3}$. Ultrafine particles dominated rooftop concentrations as indicated by an average UFP/TNC ration of 0.9. The

majority of particles in UFP size range occurred in the NUC mode.

**Table 2: Mean, median, minimum, and maximum of TNC, UFP as well as particle number concentrations of the three modes NUC, AIT, and ACC.**

| Particle number concentration (cm$^{-3}$) | TNC | UFP | NUC | AIT | ACC |
|---|---|---|---|---|---|
| Mean | 8,337 | 7,522 | 3,995 | 3,528 | 814 |
| Median | 7,300 | 6,447 | 3,136 | 3,027 | 681 |
| Minimum | 3,099 | 2,828 | 1,732 | 801 | 77 |
| Maximum | 53,879 | 48,145 | 37,442 | 30,820 | 16,741 |

### 3.4 Footprint analysis

The particle fluxes measured at our rooftop site represented a surface area of about 5.3 km² (with respect to the 80 % contour

line). The flux peak contribution, however, was from an area situated to the W/SW of the site at a distance of about 50 to





350 m (Figure 4). The peak contribution source area slightly shifted northwards in the second year and southwards in the third year. However, the source areas were very similar in the individual years. A detailed analysis of source-area weighted contribution of land-use indicates 60 % of built-up area, 25 % traffic area, 11 % green area, and 3 % water surface in the flux footprint (Figure 5). The land-use fractions were similar in the individual study years. However, it has to be taken into account

that streets vary in terms of traffic intensity, so that the strength of traffic-related sources might differ between individual years.

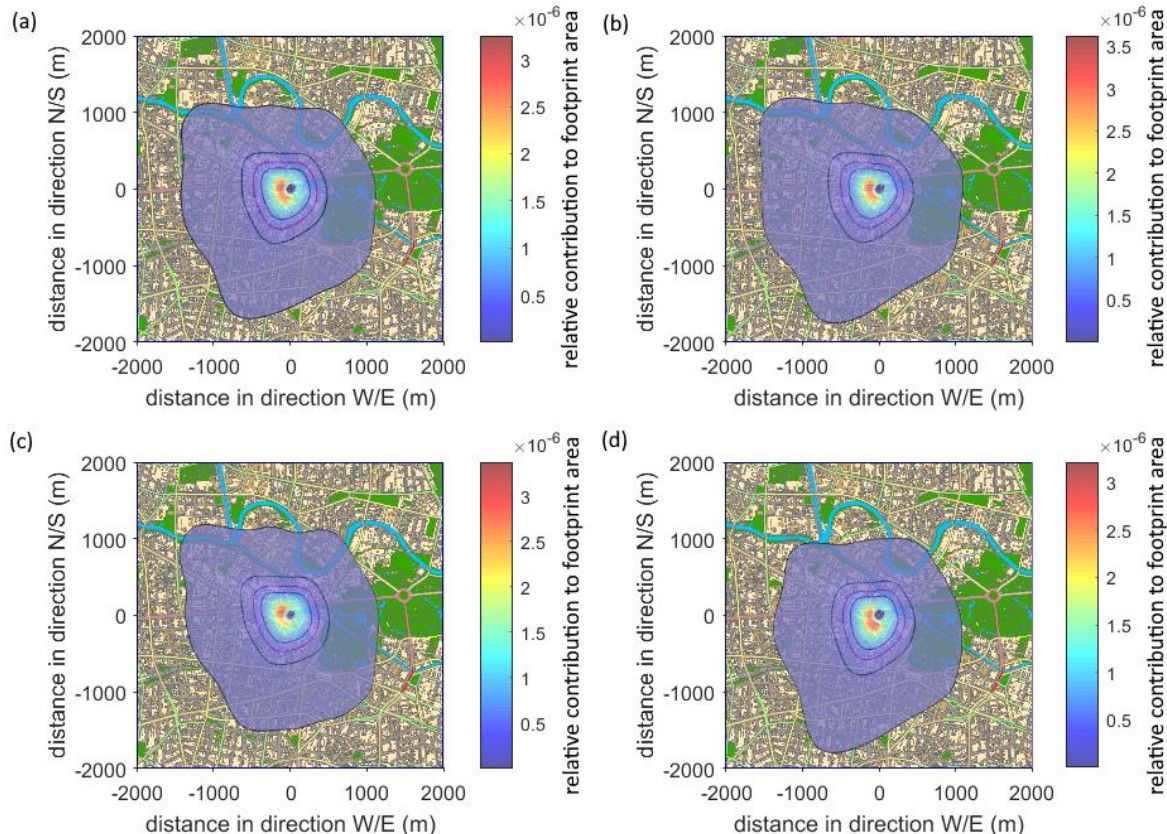

**Figure 4: Footprint climatology of (a) the entire three-year period, (b) the first year, (c) the second year, and (d) the third year. Flux footprints were calculated for a 4 km x 4 km area (4 m spatial resolution). For reasons of clarity only the 40 %, 50 % (purple), 60 %,**
**and 80 % contour lines are shown (footprint model: Kljun et al.,2015; map data sources: Geoportal Berlin, 2014 (modified), 2021; Umweltatlas Berlin, 2017).**





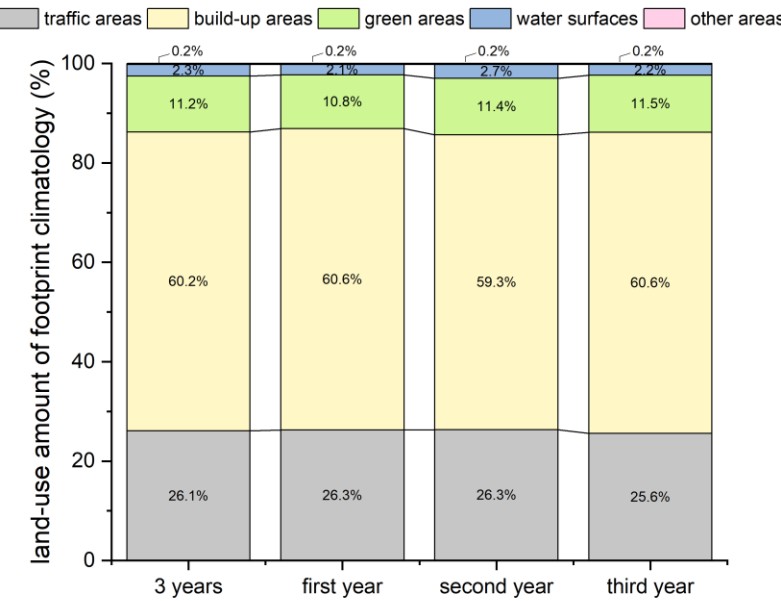

**Figure 5: Amount of land-use contributing to the flux footprint climatologies separated according to the different time periods of measurement. For data analysis, the footprint information in a 4 km x 4 km grid centred on the measurement site was used (4 m spatial resolution). This footprint area completely includes the 80 % contour line.**

### 3.5 Total and ultrafine particle number fluxes

The measured particle number fluxes indicated the study area in central Berlin as a net source of particles with a three year median $F_{TNC}$ of $0.86 \times 10^8$ m$^{-2}$ s$^{-1}$ and $0.83 \times 10^8$ m$^{-2}$ s$^{-1}$ for $F_{UFP}$ (cf. Table 3). The majority of 91.4 % of fluxes were upward (emission) fluxes, whereas 8.6 % of data showed downward (deposition) fluxes. Annual estimates of median $F_{TNC}$ and $F_{UFP}$ showed a decrease by -9.6 % (-8.9 % for $F_{UFP}$) from the first to the second year and a further -5.9 % (-6.1 % for $F_{UFP}$) from the second to the third year (Figure 6a, Table 3). We observed a limited number of frequency of higher $F_{UFP}$ (e.g. > $1.2 \times 10^8$ m$^{-2}$ s$^{-1}$) both in the second and third year (Figure 6b). The ratio of $F_{UFP}$ to $F_{TNC}$ is 0.96 whereas the average nucleation mode particle flux ($F_{NUC}$) accounted for 63 % of $F_{TNC}$ (data not shown here). As the ultrafine size range clearly dominated $F_{TNC}$, we will focus on $F_{UFP}$ in subsequent analysis.





**Table 3: Mean, median, minimum, and maximum $F_{TNC}$ and $F_{UFP}$ for the three measurement years and the four meteorological seasons spring (MAM), summer (JJA), autumn (SON), and winter (DJF).**

| $F_{TNC}$ (x $10^8$ m$^{-2}$ s$^{-1}$) | 3 years | First year | Second year | Third year | MAM | JJA | SON | DJF |
|---|---|---|---|---|---|---|---|---|
| Mean | 1.19 | 1.32 | 1.15 | 1.07 | 1.21 | 1.12 | 1.07 | 1.36 |
| Median | 0.86 | 0.94 | 0.85 | 0.80 | 0.86 | 0.82 | 0.81 | 0.97 |
| Minimum | -16.43 | -16.43 | -14.77 | -10.69 | -11.51 | -16.3 | -8.54 | -14.30 |
| Maximum | 26.78 | 22.48 | 24.88 | 26.78 | 26.78 | 24.88 | 18.38 | 22.8 |
| $F_{UFP}$ (x $10^8$ m$^{-2}$ s$^{-1}$) | 3 years | First year | Second year | Third year | MAM | JJA | SON | DJF |
| Mean | 1.14 | 1.27 | 1.10 | 1.04 | 1.17 | 1.08 | 1.03 | 1.31 |
| Median | 0.83 | 0.90 | 0.82 | 0.77 | 0.83 | 0.79 | 0.78 | 0.94 |
| Minimum | -16.51 | -16.51 | -14.74 | -10.73 | -11.53 | -16.51 | -8.57 | -14.34 |
| Maximum | 26.72 | 22.37 | 24.76 | 26.72 | 26.72 | 24.76 | 16.95 | 22.37 |

$F_{UFP}$ was characterised by distinct seasonal variation, i.e. largest average fluxes occurred in winter whereas fluxes were lower in autumn and summer (Figure 7, Table 3). Generally, a lower frequency of deposition events and higher emission fluxes prevailed during winter. However, due to consistent positive monthly median values the site footprint was a net particle emission source in every single month.

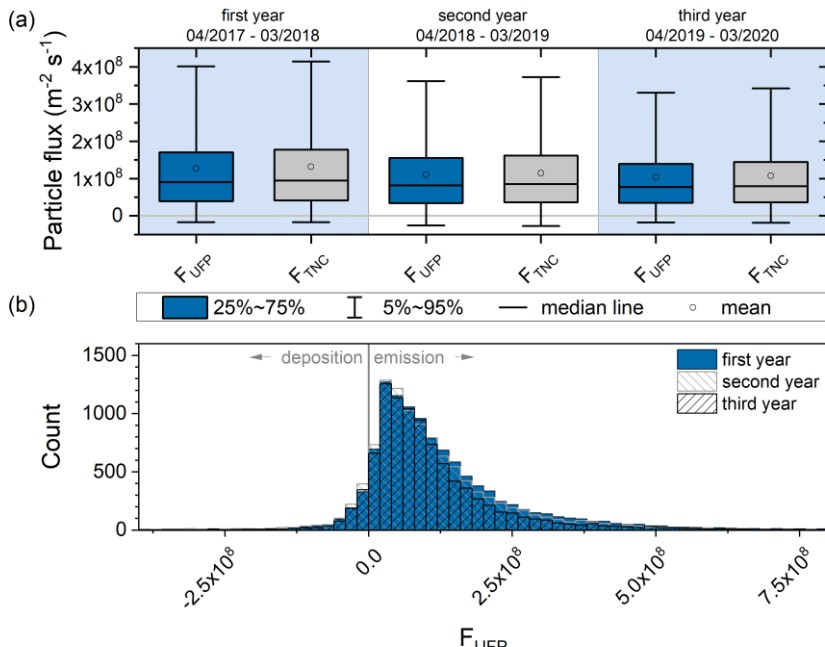

**Figure 6: (a) Range of $F_{TNC}$ and $F_{UFP}$ and (b) the frequency distributions of $F_{UFP}$ for each of the three years of measurement.**





The seasonal pattern is also evident on the mean diurnal cycle (Figure 8a). Strongest emission fluxes occurred in winter with a maximum $F_{UFP}$ of about $2.0 \times 10^8$ $m^{-2}$ $s^{-1}$, whereas lowest fluxes were observed in autumn and summer. $F_{UFP}$ increased after sunrise and showed two local maxima at around 9:30 LT and in the early afternoon (12:30 to 14:00 LT). The moderate decrease at around noon was related to reduced traffic intensity (cf. Figure 8d). Average $F_{UFP}$ varied between $0.2 \times 10^7$ $m^{-2}$ $s^{-1}$ at night

and $2.0 \times 10^8$ $m^{-2}$ $s^{-1}$ during day. The morning increase in $F_{UFP}$ coincided with an increase in atmospheric turbulence and a more unstable boundary layer. In the evening, $F_{UFP}$ decreased as atmospheric stratification changed from unstable to neutral conditions (Figure 8c).

The daily amplitude of $F_{UFP}$ clearly differed between working days and weekends (Figure 8b). Whereas working days showed a strong morning increase of $F_{UFP}$ and a large amplitude over the diurnal course, the weekend was characterised by a reduced

morning increase and only half the amplitude in comparison to working days. This difference was in accordance with the variation of traffic intensity between working days and weekends (Figure 8d).

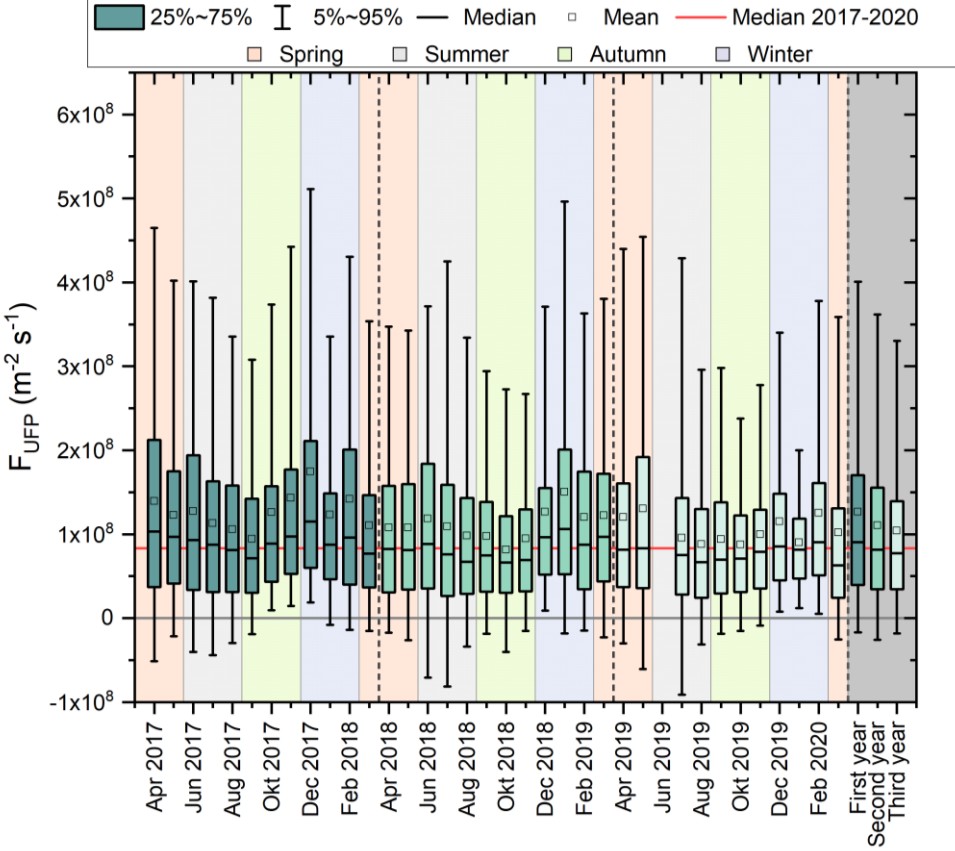

**Figure 7: Variation of monthly and annual $F_{UFP}$ during the study period.**



**Figure 8: Diurnal cycles of (a) average and seasonal $F_{UFP}$, (b) average $F_{UFP}$ for weekdays and weekends with standard deviation (SD), (c) median stability parameter $(z-d)/L$ for each season and the whole measurement period with z = measurement height, d = displacement height, and L = Monin-Obukhov length, and (d) average traffic intensity at HS and S17J.**

## 3.6 Size-resolved particle number fluxes

Size-resolved particle number fluxes were characterised by distinct temporal variation on the diurnal cycle (Figure 9a). Strongest emission prevailed in the smallest size bin ($D_p < 12$ nm) with a maximum in the early afternoon. Coarser particles ($D_p > 100$ nm), however, showed little variation regardless of the day of week. Although deposition fluxes occurred (Figure 9b), positive average fluxes were prevailing in all size ranges emphasising the dominance of net particle emission from the flux footprint. The frequency of particle deposition events peaked in the ACC mode (Figure 9b). On working days, the deposition frequency of UFP was highest during night, especially in the second half of the night. On weekends, UFP deposition



more often occurred in the morning to pre-noon hours. The timing of deposition fluxes was associated with the daily minimum in $F_{UFP}$, low traffic intensity, and a neutral or stable atmosphere (cf. Figure 8). Generally, NUC mode particles showed higher deposition probabilities than AIT mode particles. This was evident during the entire day and for every day of the week.

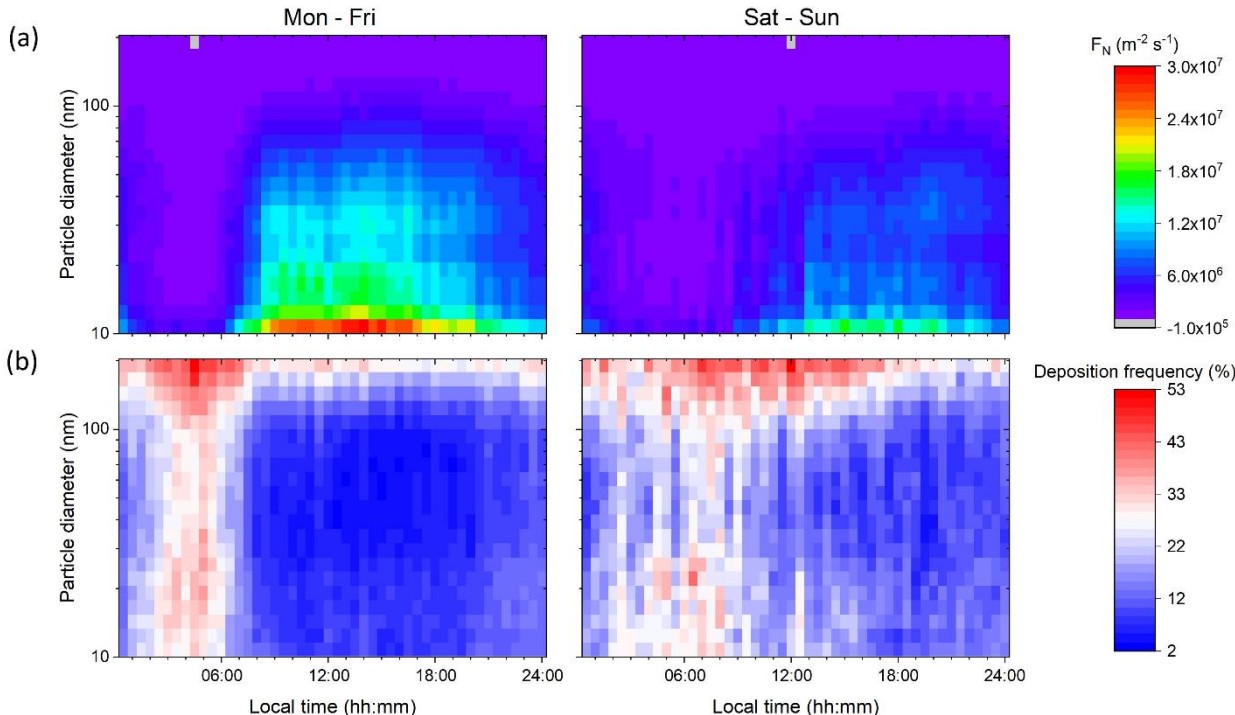

**Figure 9: Diurnal cycles for a weekday (Mon – Fri) and a day of weekend (Sat – Sun) of (a) size-resolved average particle number fluxes ($F_N$) and (b) the frequency of particle deposition. In (a) colours indicate the strength of emission fluxes while grey colour symbolises a slight particle deposition.**

On the seasonal cycle, highest emission fluxes predominantly occurred in the smallest size bins ($D_p < 30$ nm) during winter (average $F_{TNC}$: $1.44 \times 10^8$ m$^{-2}$ s$^{-1}$ and $F_{UFP}$: $1.38 \times 10^8$ m$^{-2}$ s$^{-1}$ in December, Figure 10a, b). The lowest average particle number fluxes were evident in late summer and autumn. Additionally, the frequency of particle deposition events was higher in summer/autumn with a maximum in August and September (Figure 10c). The variation in the frequency of deposition fluxes over the size spectrum pointed to the occurrence of bidirectional fluxes.

The difference between the maximum and minimum of deposition frequencies in a monthly size spectrum (max. Δ deposition frequency, Figure 10c) might be interpreted as a measure to quantify the occurrence of bidirectional fluxes. Thus, bidirectionality in particle fluxes occurred more frequently in spring, late summer and autumn than in winter. Bidirectionality was often associated with simultaneous emission fluxes in the ultrafine and deposition fluxes in the coarser particle size range. The months of May, June, and July showed less particle deposition events but higher $F_{TNC}$ and $F_{UFP}$ than the other months in spring and summer. This was probably due to the variation of wind direction and, consequently, flux source area. The average





three-year wind directions were from NNE in May, WNW in June, and NW in July. In those directions strong particle sources

such as the busy road "Straße des 17. Juni" and the traffic circle "Ernst-Reuter-Platz" were located (cf. Figure 1).

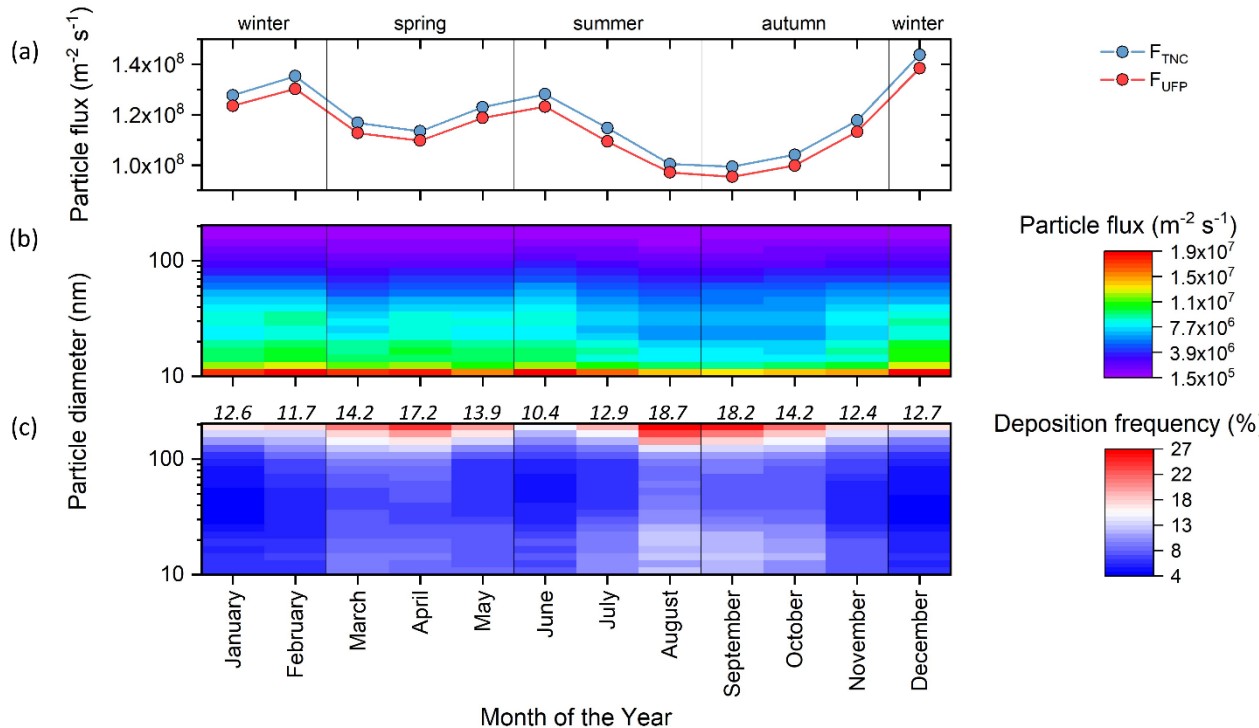

**Figure 10: Monthly variation of (a) average $F_{TNC}$ and $F_{UFP}$, (b) size-resolved average particle number fluxes, and (c) size-resolved frequency of particle deposition. In (c) the values given in italics quantify the maximum difference in deposition frequency within each monthly size spectrum (max. Δ deposition frequency, given in percentage points).**

**3.7 Wind sector analysis of modal particle number fluxes**

The wind sector-based analysis indicated net emission of particles from every wind sector (Figure 11a). The largest average $F_{TNC}$ were observed for westerly and northerly wind directions (W-N-NE), i.e. the direction in which "Straße des 17. Juni" and the traffic circle "Ernst-Reuter-Platz" are located (cf. Figure 1). In contrast, particle fluxes from easterly and southerly wind directions were smaller. Vegetated greenspaces, built-up areas but also the main road "Hardenbergstraße" were located in this

direction. For all wind directions highest particle number fluxes prevailed in NUC mode (57-72 %), followed by particles in AIT (26-38 %) and ACC modes (2-5 %, Figure 11b).

When net fluxes are distinguished into emission and deposition fluxes, the emission fluxes show similar behaviour such as the average $F_{TNC}$ due to the high frequency of emission events (cf. chapter 3.5, Figure 12a). The most frequent emission fluxes occurred for wind from 180 to 315° (Figure 12c). Strongest deposition fluxes prevailed under north-westerly wind whereas

lower deposition was evident for the remaining directions (Figure 12b). However, the most frequent deposition fluxes were





observed under winds from southerly and north-westerly directions (Figure 12d). Generally, ACC particles were more frequently characterised by deposition fluxes than other particle modes.

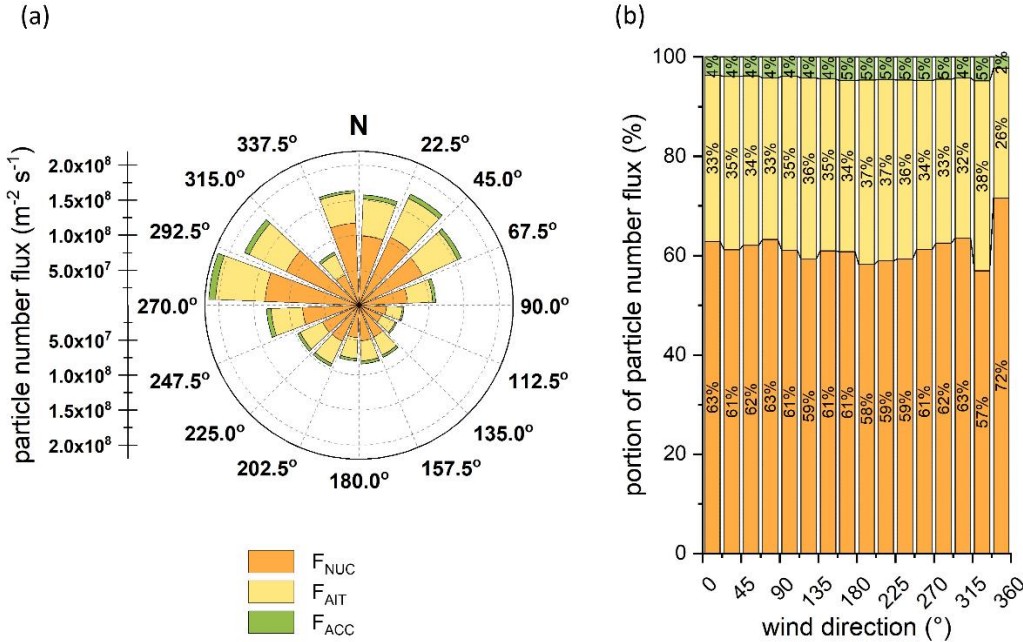

**Figure 11: (a) Average particle number fluxes and (b) fraction of specific particle mode fluxes binned into 22.5° sectors for $F_{NUC}$, $F_{AIT}$, and $F_{ACC}$.**


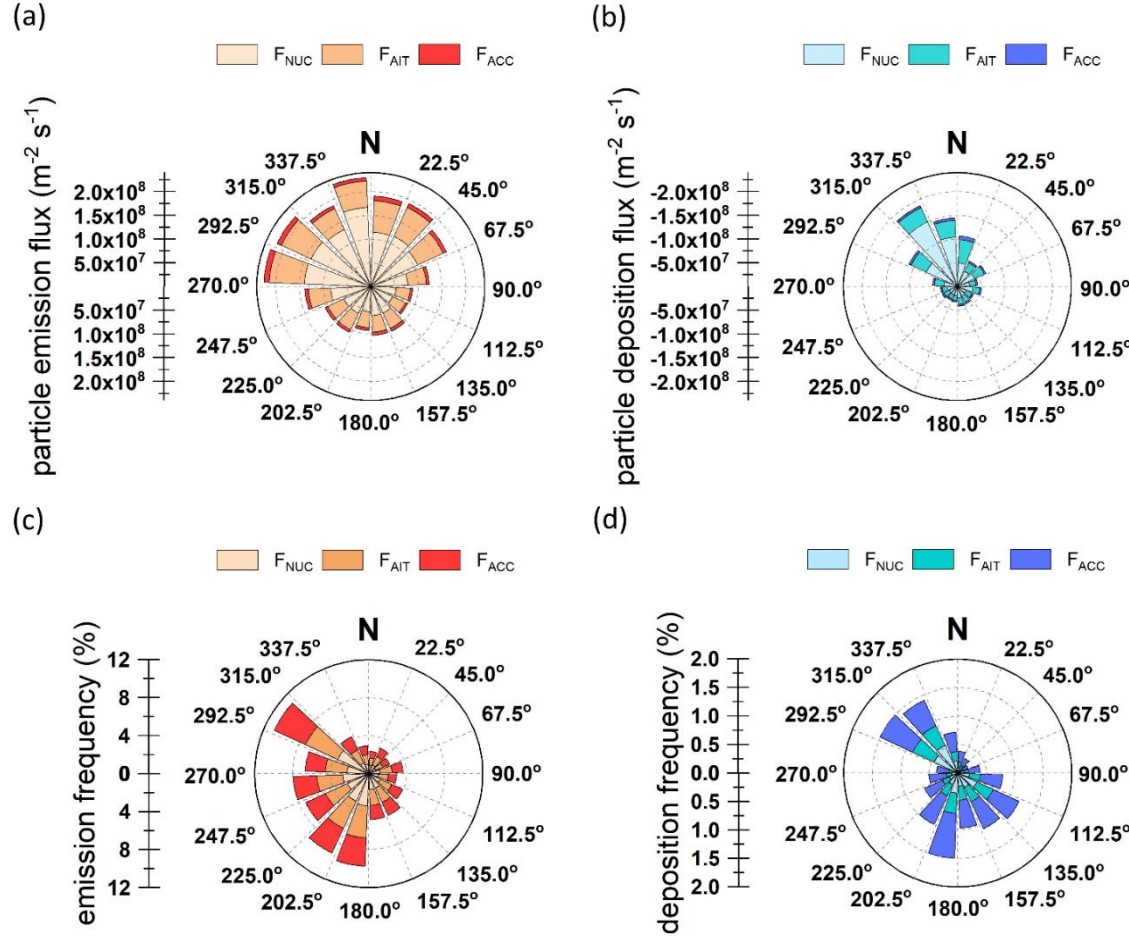

**Figure 12: Average (a) emission and (b) deposition particle number fluxes as well as the frequencies of (c) emission and (d) deposition events per wind direction sector for $F_{NUC}$, $F_{AIT}$, and $F_{ACC}$.**

## 3.8 Land-use regression analysis

The three-year data was analysed with regard to variation of land-use across the flux footprint (cf. Figure 13). We observed the sectors with the highest fraction of traffic areas to match with the highest particle fluxes, e.g. the sector with the highest traffic fraction of 53 % resulted in the largest particle emission flux with 2.05 x $10^8$ $m^{-2}$ $s^{-1}$.

To further analyse this relationship, a LUR analysis was carried out for $F_{TNC}$ and $F_{UFP}$ in the 16 wind sectors. At a confidence level of 95 %, only the land-use type "traffic areas" was highly significant with a p-value < 0.001 (Table 4).



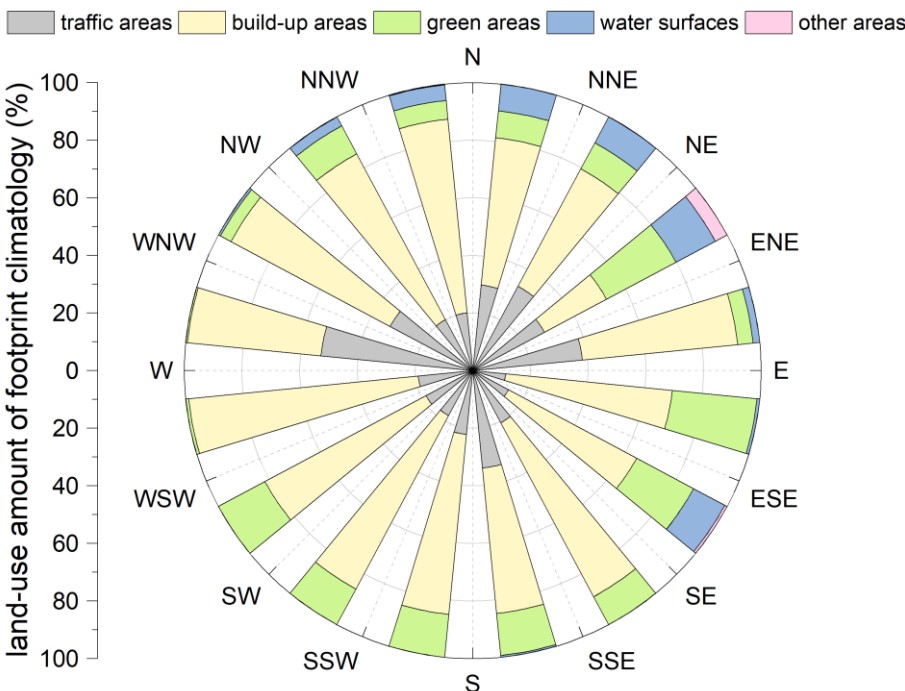

**Figure 13: Fraction of land-use per wind direction sector contributing to the three-year flux footprint. For data analysis the flux footprint was defined as a circle centred of the flux site with a radius of 2 km.**

The LUR analysis based on the three-year data set gives further evidence for the important role of traffic as UFP emission source. Traffic areas emitted $F_{UFP} = 3.44 \times 10^8$ m$^{-2}$ s$^{-1}$, followed by built-up areas with $4.87 \times 10^7$ m$^{-2}$ s$^{-1}$ (not significant, Table 4). Green areas seem to be a sink of particles with $F_{UFP} = -1.46 \times 10^7$ m$^{-2}$ s$^{-1}$ although the relationship was statistically not significant. However, the analysis does not take temporal variation of traffic intensity into account. A comparison of hourly traffic data from counting stations with particle number fluxes showed a distinct increase in $F_{UFP}$ with intensifying traffic (Figure 14). The strongest increase was obvious for $F_{NUC}$, followed by $F_{AIT}$. $F_{ACC}$ only slightly increased with traffic intensity.

**Table 4: Coefficients of the LUR analysis concerning $F_{TNC}$ and $F_{UFP}$ in the different wind sectors. Significant variables (significance level p < 0.001) are highlighted (\*).**

| | $F_{TNC}$ (m$^{-2}$ s$^{-1}$) | Standard error $F_{TNC}$ (m$^{-2}$ s$^{-1}$) | P-value | Lower 95 % (m$^{-2}$ s$^{-1}$) | Upper 95 % (m$^{-2}$ s$^{-1}$) |
|---|---|---|---|---|---|
| **Intersect** | 0 | - | - | - | - |
| **Green areas** | $-1.58 \times 10^7$ | $9.69 \times 10^7$ | 0.87 | $-2.25 \times 10^8$ | $1.94 \times 10^8$ |
| **Built-up areas** | $5.07 \times 10^7$ | $3.73 \times 10^7$ | 0.20 | $-2.98 \times 10^7$ | $1.31 \times 10^8$ |
| **Traffic area** | $3.59 \times 10^8$ | $6.87 \times 10^7$ | $0.16 \times 10^{-3}$ \* | $2.10 \times 10^8$ | $5.08 \times 10^8$ |
| **Adjusted R²** | | | | | 0.85 |





|  | $F_{UFP}$ (m$^{-2}$ s$^{-1}$) | Standard error $F_{UFP}$ (m$^{-2}$ s$^{-1}$) | P-value | Lower 95 % (m$^{-2}$ s$^{-1}$) | Upper 95 % (m$^{-2}$ s$^{-1}$) |
|---|---|---|---|---|---|
| **Intersect** | 0 | - | - | - | - |
| **Green areas** | $-1.46 \times 10^7$ | $9.44 \times 10^7$ | 0.88 | $-2.18 \times 10^8$ | $1.89 \times 10^8$ |
| **Built-up areas** | $4.87 \times 10^7$ | $3.63 \times 10^7$ | 0.20 | $-2.97 \times 10^7$ | $1.27 \times 10^8$ |
| **Traffic area** | $3.44 \times 10^8$ | $6.69 \times 10^7$ | $0.19 \times 10^{-3}$ * | $1.99 \times 10^8$ | $4.88 \times 10^8$ |
| **Adjusted R²** |  |  |  |  | 0.84 |

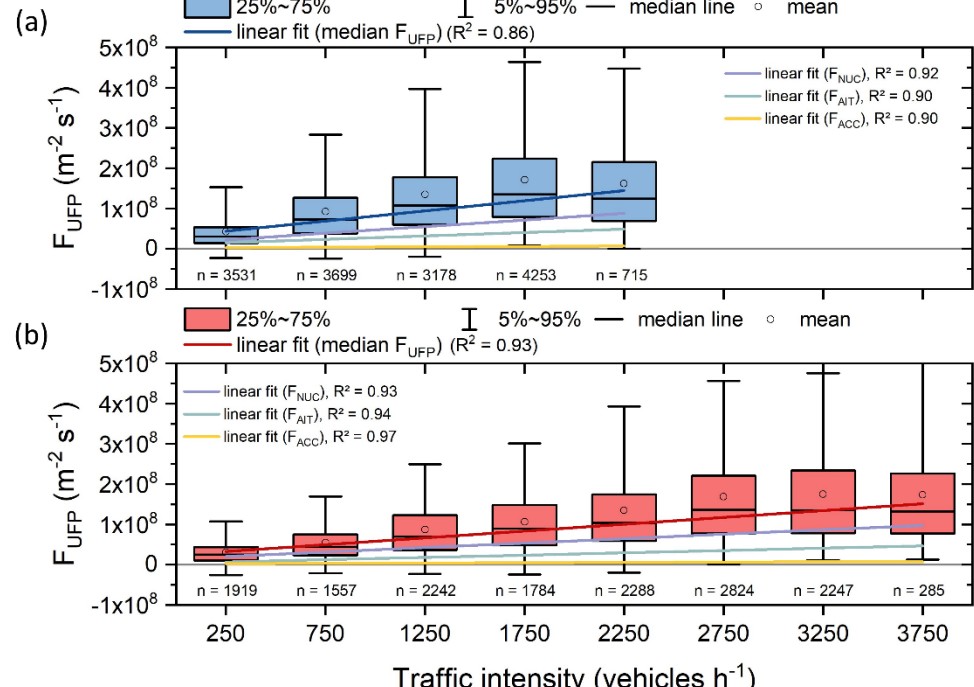

**Figure 14: Relationship between $F_{UFP}$ and traffic intensity at (a) "Hardenbergstraße" (HS) and (b) "Straße des 17. Juni" (S17J). Linear regression lines for the particle mode fluxes $F_{NUC}$, $F_{AIT}$, and $F_{ACC}$ are given.**

## 4. Discussion

The particle number fluxes observed at the Berlin site show dominance of particle emission with a three-year median $F_{TNC}$ of

$0.86 \times 10^8$ m$^{-2}$ s$^{-1}$ and $F_{UFP}$ of $0.83 \times 10^8$ m$^{-2}$ s$^{-1}$ which makes the site a net emission source of ultrafine particles to the urban

boundary layer. Urban particle number flux measurements from different locations are, however, not easy to compare given

the different diameter cut-offs of particle counting instruments. Due to the dominating impact of UFP on the number flux, the

lower cut-off is of particular importance. However, our results coincide with other studies from cities across Europe using a

similar lower cut-off in the particle counting instruments (cf. Table 1). Total number fluxes of the studies in Stockholm

(Mårtensson et al., 2006), Innsbruck (Deventer et al., 2018; Heyden et al., 2018) or Lecce (Conte et al., 2018) are in a similar

range.





Although the three observation years point to constant emission of ultrafine particles from the urban surface, annual estimates show a median $F_{UFP}$ decrease of -8.9 % from the first to the second and -6.1 % from the second to the third observation year. The observed decrease might be due to several reasons. Analysing the temporal variation of traffic count data, we found a slight reduction in the third year (Figure A3). This could be one reason for the lower median $F_{UFP}$ in the third observation year.

In addition, the decrease might be associated to the slight spatial shift of the peak contribution location in the flux footprint towards southerly directions. These source areas are characterised by a lower fraction of traffic areas and reduced traffic intensity in comparison to northerly directions. In the second year, the average wind direction and also the peak contribution location in the flux footprint shifted towards north-westerly directions. These were related to stronger particle deposition fluxes (cf. Figure 12) that tend to reduce $F_{UFP}$. Additionally, a transition of the vehicle fleet towards a higher share of low-emission

standards or electric drive might play a role. According to official car registration data, the proportion of electric vehicles in Berlin increased by a factor of three in the time period from 2017 to 2020 (Kraftfahrt-Bundesamt, 2021). Additionally, in the third observation year lower traffic intensity due to the COVID-19 lockdown restrictions starting in mid-March 2020 might have influenced our measurements and may explain the local minimum fluxes in March 2020 (cf. Figure 7). However, an in-depth analysis of effects related to reduced traffic intensity during the COVID-19 lockdown is beyond the scope of this study

but will be addressed in future work.

The analysis of size-resolved particle number fluxes points to the dominance of emission fluxes over the diurnal cycle. The strongest emission fluxes occur during daytime with a maximum of smallest particles in the early afternoon (e.g. Schmidt and Klemm, 2008; Deventer et al., 2018). The diurnal courses of particle fluxes coincided with traffic intensity demonstrating strong influence of traffic on $F_{NUC}$. The delayed increase and reduced amplitude of traffic intensity weekends compared to a

working day was clearly evident in particle flux data. At this time of low traffic intensity, particle deposition occurred much more frequently. While coarse particles ($D_p > 180$ nm) were deposited with a frequency > 20 % at any time throughout the day, UFP deposition preferably occurred in the second half of the night or other low-emission periods, e.g. weekend mornings. Simultaneous occurrence of upward and downward directed fluxes, i.e. bidirectional fluxes, was recently reported from other cities (Schmidt and Klemm, 2008; Deventer et al., 2013, 2015, 2018). A tipping point, that defines the particle diameter

separating average emission from average deposition fluxes, was reported for $160 < D_p < 190$ nm (Deventer et al., 2013, 2015). On the mean diurnal cycle no such tipping point can be identified at the present site, since our data does not show simultaneously occurring average upward and downward fluxes across the size spectrum.

We found a dominating contribution of $F_{UFP}$ to the total flux (96 %) which corresponds to findings from Innsbruck (99 %; size range: 6 nm $< D_p <$ 637 nm; Deventer et al., 2018) and London (Harrison et al., 2012). In addition, the diurnal course of $F_{UFP}$

showing a nocturnal minimum and an early afternoon maximum that is driven by traffic intensity and atmospheric stability is comparable to other findings (Dorsey et al., 2002; Deventer et al., 2013; Conte et al., 2018). A typical double rush-hour pattern of traffic intensity that is often not occurring in particle flux data (Dorsey et al., 2002; Järvi et al., 2009; Martin et al., 2009; Deventer et al., 2015) is slightly evident in the diurnal cycle of average $F_{UFP}$ in Berlin. Also the differences in the diurnal cycles of $F_{UFP}$ between working and weekend days point to dominating influence of traffic on the measured fluxes at our site. This



was confirmed by wind-dependent analysis of flux variation, the regression of fluxes against land-use in the flux source area (LUR analysis) and the temporal variation with traffic data from counting stations. Similar findings of higher fluxes from traffic areas and lower fluxes from areas with higher amounts of green surfaces and buildings were reported by Järvi et al. (2009), Ripamonti et al. (2013), and Mårtensson et al. (2006).

In each wind direction sector, we found the highest particle number fluxes in nucleation mode ($F_{NUC} \geq 57$ % of average $F_{TNC}$,

cf. Figure 11b). Road traffic is the main source of particles in the size range below 50 nm (Morawska et al., 2008). Especially rush-hour periods are characterised by a maximum of traffic-related particle emission in the size range between 20 and 30 nm (Wiedensohler et al., 2002). Thus, the large contribution of $F_{NUC}$ confirms the high influence of traffic intensity on fluxes at this site.

The deposition frequency seems to be related to the strength of local sources as indicated by the increase of deposition

frequency with decreasing $F_{UFP}$ and $F_{TNC}$ (cf. Figure 10). In addition, the occurrence of bidirectional fluxes in the annual course also seems to depend on the strength of the surrounding sources with a more frequent appearance in seasons with lower emissions. Thus, they occur more often in summer and autumn than in winter (cf. Figure 10c). Deventer et al. (2015) also reported differences in particle deposition between summer and winter and found an increase of deposition fluxes for $D_p >$ 170 nm in spring and summer.

**5 Conclusions**

We report on the first multi-annual data set of urban size-resolved particle number fluxes that were measured over the size range $10 < D_p < 200$ nm from April 2017 to March 2020 in an urban area of Berlin, Germany, using the eddy-covariance technique. The Berlin site was a net source of particles with a majority of ultrafine particle emission as indicated by a $F_{UFP}/F_{TNC}$ ratio of 0.96. The magnitude and temporal variation of $F_{TNC}$ and $F_{UFP}$ were rather similar in the individual observation years,

however, a reduction of median $F_{UFP}$ of -8.9 % and -6.1 % for the second and third years was evident. This might be related to variation in flux source area, a modification in the vehicle fleet, and an effect of reduced traffic intensity in the third measurement year.

The results clearly point to traffic as the dominant influence on ultrafine particle number fluxes at the Berlin site. Due to the health effects associated with UFP, measures that result in reduced traffic intensity, e.g. by promoting alternative urban

mobility concepts, or a transition of the traffic fleet towards a higher share of electric or low-emission vehicles may help to reduce personal exposure towards UFP. Long-term observations of size-resolved particle number fluxes form different urban environments would be an important tool to monitor and asses the success of different mobility strategies and reduction measures.






**Appendix A**

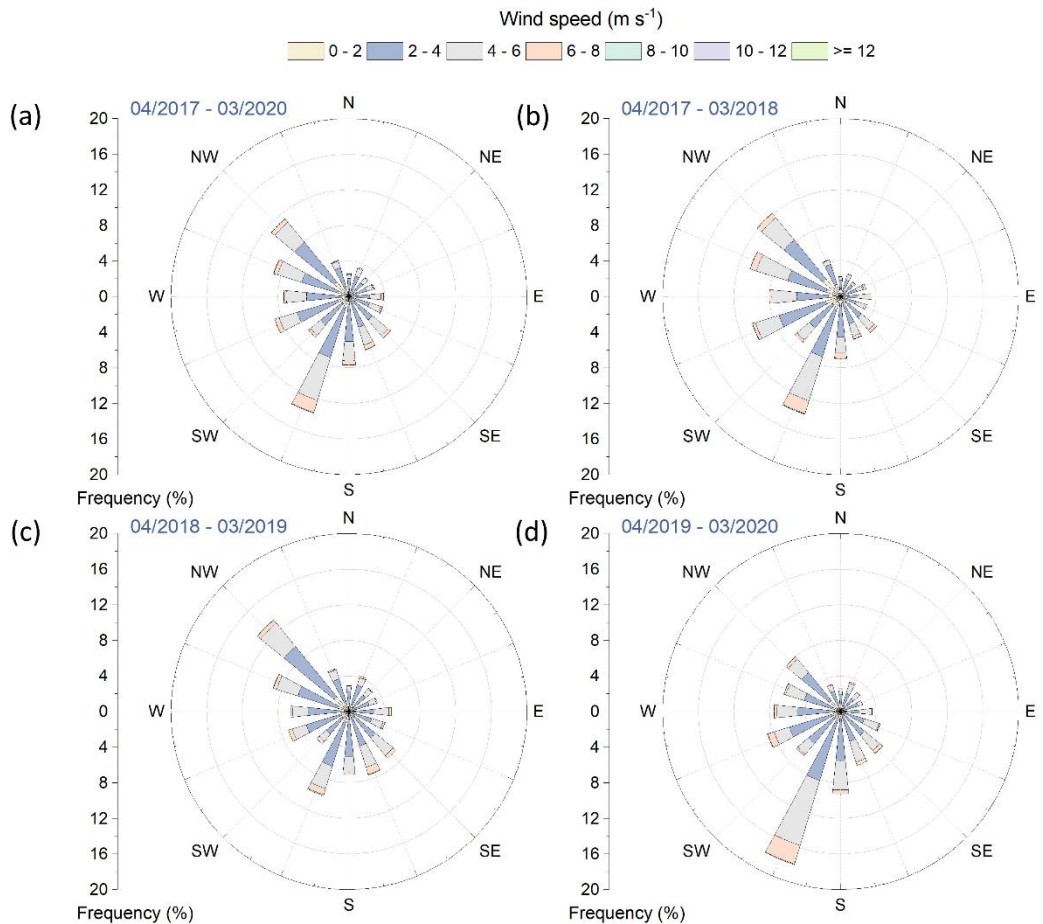

**Figure A1: Wind roses for (a) the whole three-year period, (b) the first year, (c) the second year, and (d) the third year.**





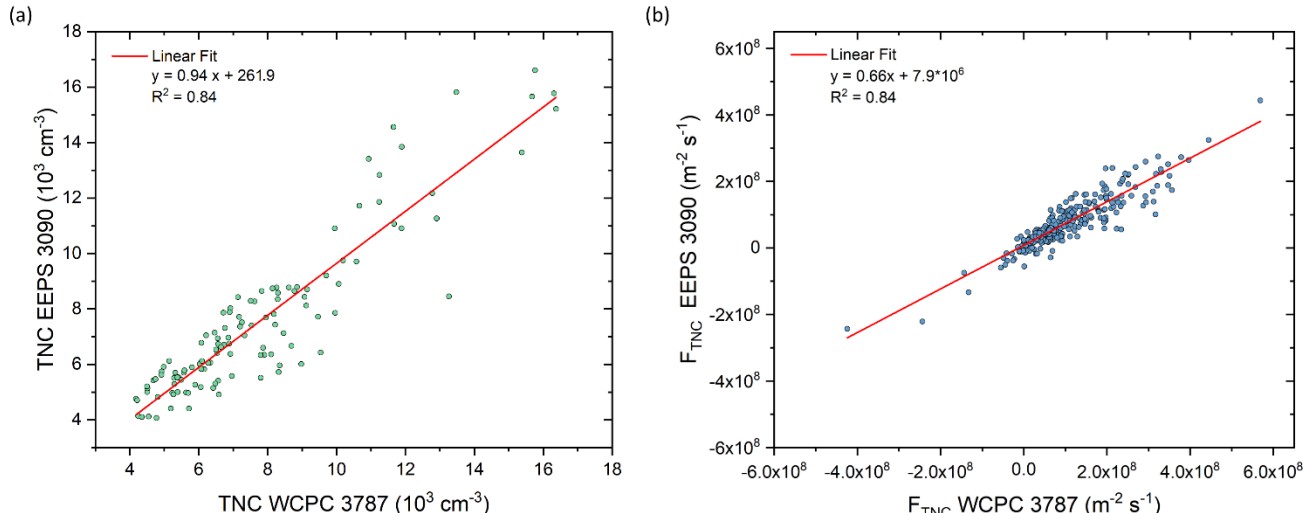

**Figure A2: Measurement comparison of EEPS 3090 with WCPC 3787 at the Berlin site concerning (a) total particle number concentration (TNC, 4 days, July 2020) and (b) total particle number flux ($F_{TNC}$, 12 days, September 2020). The data of EEPS relate to particle sizes between 10 and 200 nm whereas the WCPC counts particles with $D_p > 5$ nm.**

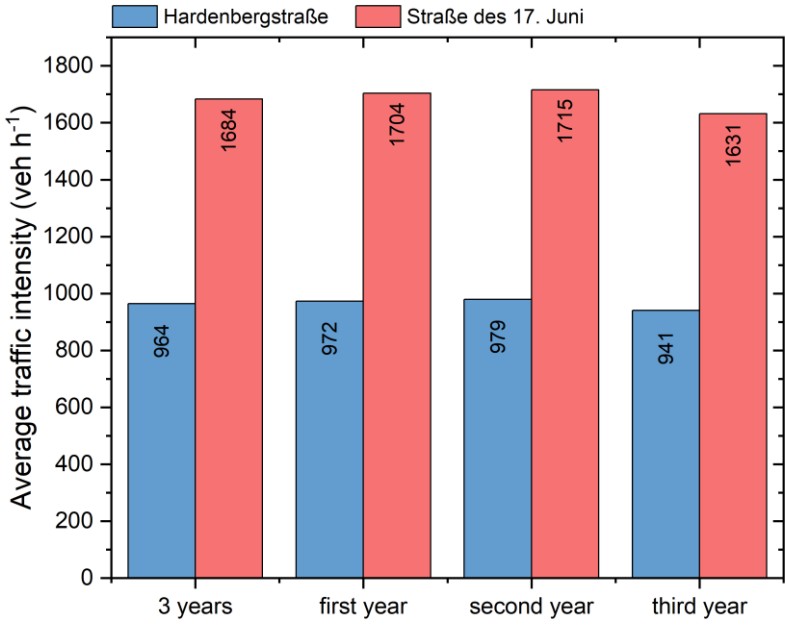

**Figure A3: Average traffic intensity measured at the two traffic counting stations "Hardenbergstraße" (HS) and "Straße des 17. Juni" (S17J) for the entire measurement period and each observation year.**





**Data availability**

The data used in this study are publicly available: "Three years of size-resolved eddy-covariance particle number flux measurements in Berlin, Germany" on https://publikationsserver.tu-braunschweig.de/receive/dbbs_mods_00069675.

**Author contribution**

SW acted as the supervisor with responsibilities concerning conceptualisation, funding acquisition, project administration as well as reviewing and editing of the paper draft. Together with AS, he designed the experiments. AS took care of data curation, maintaining the flux site, formal data analysis, measurement operation and data collection, methodology, software use and development for data analysis as well as data visualisation. Additionally, AS prepared the original draft of this paper.

**Competing interests**

The authors declare that they have no conflict of interest.

**Acknowledgements**

This study was supported by the German Federal Ministry of Education and Research (BMBF) under Grants FKZ 01 LP 1602 D (Urban Climate under Change, Module 3DO) and FKZ 01 LP 1912 D (Urban Climate under Change, Phase II, Module
3DO+M). We would like to thank Andreas Müer from VMZ Berlin Betreibergesellschaft mbH for providing traffic data from two counting stations. Furthermore, we would like to thank our project partners (Fred Meier, Dieter Scherer) of the TU Berlin, Institute for Ecology, Chair of Climatology, for providing hourly measurement data of mixing layer height for flux footprint prediction and for being able to install sensors at their site. We thank Hagen Mittendorf, Technische Universität Braunschweig, for his support in setting up and maintaining the flux site.

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
