# Peer review of "Measurement report: Three years of size-resolved eddy-covariance particle number flux measurements in an urban environment"

_Atmospheric Chemistry and Physics, 2021_

## Author Comment (AC1)

**Replies to the Referees' comments**

We would like to thank the referees for their interest in our work and the helpful comments and questions to our manuscript "Measurement report: Three years of size-resolved eddy-covariance particle number flux measurements in an urban environment". Herewith, we would like to respond to the comments, which are set in *italics*. Our responses are written in blue.

**Anonymous Referee #1:**

*GENERAL COMMENTS:*

*The paper presents a report on 3-years measurements of particle number flux*
*measurements in the city of Berlin from April 2017 to March 2020. It is a long term monitoring campaign focused on fluxes of ultrafine particles and accumulation mode particles (10 nm > Dp < 200 nm). The paper presents average diurnal, seasonal and annual fluxes in order to find a correlation with diurnal concentration peaks, seasonal emission sources and the relationship between FTNC (total number particle flux) and FUFP (ultrafine particle number flux) with land use in the flux footprint. The paper is very well structured, clear and detailed. The methodology is appropriate and supported by a wide bibliography. The authors show that the urban domain investigated is a net source of particles (e.g. particles emissions prevail over particles deposition, especially with reference to ultrafine particles) and the presented results are aligned with other urban case studies. Particles fluxes are investigated considering meteorological parameters of the specific monitoring period (like precipitation, wind speed, wind direction, temperature), the diurnal atmospheric stability classes but even the land use of the territorial domain and hourly traffic intensity data supplied by two different traffic counting stations located near the monitoring site. The authors stress the strong relationship between particles fluxes and the traffic intensity of specific roads in Berlin and exclude a significant contribution of green areas and/or built areas nearby. I would suggest, as future development of the research, to add a multistage cascade impactor to be used as gravimetric analyser in order to get a size-resolved chemical speciation of particles for a source apportionment study. In my opinion the paper only needs minor revisions before publication, clarifying some assumptions and to better support the conclusions of the work.*

We agree that size-resolved chemical particle speciation could be an important benefit to conduct a more detailed source apportionment and process analysis at the present site. However, one disadvantage comes with a lower temporal resolution of the data as the particle mass for chemical analysis would probably have to be collected over a longer time-period compared to the 30 min eddy covariance data. However, chemical analysis could certainly be a helpful addition to the particle flux data and we will keep that in mind for future research at this site.

*SPECIFIC ISSUES:*

1) *In the paper it could be interesting to better explain which types of "green areas" are present in the territorial domain as possible emitters or sinks of PM. Maybe the different magnitude of $F_{NTC}$ and $F_{UFP}$ over the years and seasons is not only due to a change in traffic intensity or wind direction*
We further classified "green area" into more specific land-use types such as forests, bushes/single trees, grass with and without single trees, and gardens. This information is available from a habitat type mapping provided by Geoportal Berlin (2014). We added this information to the map in Figure 2 (cf. Figure R1). Unfortunately, more detailed information on (other) types of green areas is not available.
It is correct that there may be an influence of the vegetated surfaces on particle fluxes, however, the land-use regression analysis (cf. section 3.8) indicated that the land-use type "green area" did not show a statistically significant relation to the particle number fluxes. Hence, we suppose the impact of green areas to be of minor importance in comparison to traffic areas/intensity.

[Figure]

**Figure R1:** Eddy covariance measurement site in central Berlin near Ernst-Reuter-Platz (ERP; data sources: Geoportal Berlin, 2014 (modified), 2021; Umweltatlas Berlin, 2017).

2) *Considering traffic intensity data, are there more specific data available? In Figure A3 the differences in traffic intensity among the 3 years investigated are not statistically significative to justify a change in urban FUFP Maybe data more strictly related to the type of vehicles running in Berlin during the 3 years could be more representative to support the conclusions (changes in the vehicles fleet, lower traffic intensity in 2020 due to COVID pandemic, etc…).*

You are right, more specific traffic data would certainly be helpful to analyse annual differences in particle number flux in more detail. Unfortunately, we do not have access to more precise traffic information such as the vehicle fleet composition. However, it should be emphasized that the main reason for annual variation of particle fluxes is most likely due to the fact that traffic areas vary in their traffic intensity, i.e. major vs. minor roads. Thus, shifts in the flux footprint might result in a similar proportion of traffic area, but a higher traffic intensity due to a larger fraction of major roads. In addition, higher deposition fluxes in some wind sectors can also affect the annual mean/median fluxes if the wind direction distribution changes.

To quantify this impact, we calculated a flux footprint-weighted average daily traffic intensity (ADT, data source: Umweltatlas, 2017) for each year (Figure R2). The ADT was footprint-weighted to calculate the specific traffic intensity that drives particle number fluxes. The analysis demonstrates that the highest ADT occurred in the first year, and lower values in the second and third year. This is very similar to the variation of particle number fluxes (Figure 6 and Table 3 in the manuscript), which demonstrates that the traffic intensity is strongly correlated to particle number fluxes and probably the main reason for particle flux variation.

We added the following Figure R2 (as Figure A3a) and modifications to the revised manuscript:
**L124 (Section 2.3 "Data handling"):**
"To analyse the impact of varying amounts of major and minor roads in the annual flux footprints, we calculated a footprint-weighted average daily traffic intensity (ADT). For this, the ADT shape file (data source: Umweltatlas, 2017) was converted into a raster grid (4 km x 4 km with 4 m spatial resolution, comparable to the footprint climatology raster) and weighted with the footprint climatology to calculate the ADT."
**L344 (Section 4, "Discussion"):**
"As traffic areas in the flux footprint have different traffic intensity (i.e. major and minor roads), the annual variation of surface fractions of traffic areas in the flux footprints might not be sufficient to explain

particle number flux differences. However, the variation of traffic impact can be estimated using a footprint-weighted ADT, which estimates annual variation of traffic influence by quantifying the amount of major and minor roads contributing to the annual flux footprint (Figure A3a). This indicates a relationship between particle fluxes and footprint-weighted traffic contribution, which we assume to be the major reason for the observed variation in annual average particle fluxes."

[Figure]

**Figure R2:** Footprint-weighted average daily traffic intensity (ADT) for each observation year (data source: Umweltatlas Berlin, 2017; footprint model: Kljun et al., 2015).

3) *The article only presents a table with mean, median, min and max concentrations of TNC, UFP as well as NUC, AIT and ACC mode particles (Table 2). The maximum concentration values indicated (very high!) should be justified. It could be interesting to present some graphs with the particles size distribution from 10 nm to 200 nm and the seasonal diurnal courses of the 3 modes (NUC, AIT, ACC) considering the median values to better correlate the fluxes with potential local emission sources.*

Thank you for that comment. We improved the characterisation of the measurement site for particle number concentrations by adding two more values of descriptive statistics, i.e. the 2 and 98 percentile (cf. Table 2, Table R1). The absolute maximum 30 min values are distinctly larger compared to the 98 %-value (e.g. a factor 2.5 for TNC), but plausible at a traffic exposed site with traffic emission from a busy 6-lane road with about 37,300 vehicles day$^{-1}$ (Umweltatlas Berlin, 2017), even if the sample inlet is located at 57 m above ground level.

In the present study, we focussed on size-resolved particle number fluxes to investigate local emission sources and sinks at the site whereas number concentrations were highlighted in Table 2 to characterise ambient conditions of the site, e.g. a 'traffic-exposed rooftop site'. With regard to the drivers of particle fluxes, we do not see benefits in taking a closer look at number concentrations, since the particle fluxes give the most direct information about sources and sinks.

**Table R1:** Statistical quantities of TNC, UFP as well as particle number concentrations of the three modes NUC, AIT, and ACC.

| Particle number concentration (cm$^{-3}$) | TNC | UFP | NUC | AIT | ACC |
|---|---|---|---|---|---|
| Minimum | 3,099 | 2,828 | 1,732 | 801 | 77 |
| 2-percentile | 3,794 | 3,488 | 1,986 | 1,126 | 177 |
| Median | 7,300 | 6,447 | 3,136 | 3,027 | 681 |
| Average | 8,337 | 7,522 | 3,995 | 3,528 | 814 |
| 98-percentile | 19,947 | 18,734 | 12,725 | 9,500 | 2,287 |
| Maximum | 53,879 | 48,145 | 37,442 | 30,820 | 16,741 |

4) *Figure 5 does not show significant differences in land use over the 3 years investigated so that the authors recall a change in traffic intensity during the years, not justified by traffic intensity data reported in Figure A3. Please support better your assumptions.*

It is correct that we argue traffic intensity to be the main reason for variation in particle number flux. The data from the traffic counting stations (cf. Figure A3) may explain the reduced traffic intensity in the third year as a reason for the lower particle number fluxes during that year. However, another reason for annual flux variation is introduced by a spatial shift of the peak contribution location in the flux footprint. Even if the footprint climatologies of the three years show a similar proportion of traffic area, the traffic intensity is not necessarily the same (i.e. different types of roads). A more detailed analysis of the footprint-weighted average daily traffic intensity (cf. Comment 2, Figure R2) supports this argument.

5) *Considering data availability I suggest to clarify better the missing periods over the 3 years since potentially some seasonal flux trends could be affected by a considerable lack of data*
Thanks for this comment. We analysed the temporal variation of data availability and plotted the seasonal average diurnal cycle of $F_{UFP}$ data availability (Figure R3). It is obvious that data availability is generally lower during night (higher rejection of EC data due to quality flags as is common in EC measurements due to reduced turbulence and stronger stability) and that summer and spring are characterised by a lower data availability due to the 2-month gap in observation in 2019 (due to instrument inter-comparison).

To clarify this issue, we added the following sentences to the revised manuscript.

**L131 (Section 2.4, "Data availability"):**
"Data availability was generally lower during night (higher rejection of EC data due to quality flags as is common in EC measurements due to reduced turbulence and stronger stability) and showed highest average diurnal data availability for $F_{UFP}$ in autumn (69 %), followed by winter (58 %), spring (56 %) and summer (54 %)."

[Figure]

**Figure R3:** Average diurnal cycle of $F_{UFP}$ data availability for an average season of the three-year period.

6) *Deposition fluxes should be justified better (lines 295-297). Why were deposition fluxes observed mainly with southerly and north-westerly winds according to the authors? And why mainly ACC particles were deposited?*
We revised the paragraphs as follows to clarify these points.

**L295 (Section 3.7, "Wind sector analysis of modal particle number fluxes"):**
"However, the most frequent deposition fluxes were observed under winds from southerly and north-westerly directions (Figure 12d). Since particle deposition occurred most often during night (cf. Figure 9b), the wind direction related frequency distribution of deposition fluxes (Figure 12d) was similar to the night time wind rose (not shown here) which was characterised by most frequent winds from southerly and north-westerly directions. The wind direction related frequency distribution of emission fluxes, however, was similar to the daytime wind rose (not shown here). Generally, ACC particles were more frequently characterised by deposition fluxes than other particle modes."

**L 379 (Section 4, "Discussion"):**
"The deposition frequency seems to be related to the strength of local sources as indicated by the increase of deposition frequency with decreasing $F_{UFP}$ and $F_{TNC}$ (cf. Figure 10). This may also be the reason for the higher frequency of particle deposition in ACC than UFP, since ACC emission fluxes are significantly lower than AIT and NUC mode fluxes (cf. Figure 9a)."

7) *Lines 333-335: Please specify the ranges of total number fluxes coming from the literature studies compared to the paper results. The sentence is too general.*
Thanks for that comment. We revised the sentence as follows:

**L333 (Section 4, "Discussion"):**
"Average total number fluxes in Stockholm (Mårtensson et al., 2006), Innsbruck (Deventer et al., 2018; Heyden et al., 2018) and Lecce (Conte et al., 2018) are in the same order of magnitude varying between 0.5 and 3.0 x $10^8$ $m^{-2}$ $s^{-1}$."

8) *An additional local emission source of particles fluxes in the winter season can be potentially given by domestic heating in an urban area. Have you excluded this potential source coming from the build up areas considering the land use impact on seasonal particles fluxes?*
You are correct that there are further urban particle sources such as domestic heating during winter. We added this short paragraph referring to this aspect:

**L384 (Section 4, "Discussion"):**
"The highest particle number fluxes were observed in winter (cf. Figure 10a) due to a larger number of emission sources (e.g. domestic heating) and reduced atmospheric dilution. However, the LUR analysis illustrates that the relationship between sources from built-up areas and particle number fluxes is statistically not significant and that the emission fluxes from built-up areas are one order of magnitude smaller than traffic area fluxes (cf. Table 4). Thus, other local emission sources such as domestic heating in winter are of limited influence for particle number fluxes at this site in contrast to traffic related sources."

*TECHNICAL CORRECTIONS:*

*The following type errors need to be corrected in the paper:*
9) *Line 152: eliminate "not"*
In contrast to the usual procedure in land-use regression modelling, in which non-significant variables are discarded, they were not eliminated in this analysis to illustrate the sign and magnitude of the relationship between land-use and particle number fluxes. Our approach at this point was not to create the best performing LUR model for purposes such as forecasting particle fluxes. Thus, the sentence is correct and the word "not" should not be eliminated.
10) *Line 192: "ratio" instead of "ration"*
Done.
11) *Line 306: "data set" instead of "data"*
Done.
12) *Caption of Figure 13: "centered on" instead on "centered of"*
Done.

**Anonymous Referee #2:**

*GENERAL COMMENTS:*

*This paper presents a unique long-term dataset of particle flux measurements between 10 nm and 200 nm. Findings indicate that the city is a net source of particles with minimal deposition occurring within the flux footprint and study duration. The duration of these measurements allows the authors to observe changes in flux over a variety of temporal scales: annual, seasonal, weekday/weekend, and daily. Wind sectors associated with higher traffic intensity are shown to have a larger upward flux than sectors containing lower traffic or green spaces. The structure of the paper is excellent and follows a very clear methodology of flux analysis. Complimentary measurements such as aerosol characterization and source apportionment techniques would be helpful to better understand the drivers of the observed fluxes and wind sector differences. I find this paper to be a good contribution to the existing literature. Only minor revisions are required before publication.*

*SPECIFIC COMMENTS:*

1) *L37 – Diel instead of diurnal might be more appropriate to indicate that the full 24 hour day is being referenced, not just the daytime hours.*
   Thanks for your comment, however, we would like to stick to the term 'diurnal' whenever we refer to the daily cycle and temporal variation on the daily cycle. If we refer to a quantity that was analysed with regard to a specific period during daytime hours (e.g. 10:00 to 16:00 LT), we use the term 'daytime' instead.

2) *L73 – Were the data logged on the same computer? If they were logged separately how were the computers clock-synced?*
   Yes, the data of the particle spectrometer and the ultrasonic anemometer were synchronously logged on the same computer.

3) *L78 – I am a little confused by the description of the sampling flow. Is the EEPS sampling at 10 L/min or is the bypass line sampling at 10 L/min? If it is the latter, I'm concerned that the laminar flow through the bypass line will introduce bias to the particle size distributions and lag time. Sample flow through the bypass line should be turbulent. Particle attenuation through the bypass line could be a substantial factor impacting these data.*
   Yes, the EEPS did sample with a flow rate of 10 L/min and was connected to the sampling line/inlet so that a sampling flow rate of approx. 10 L/min can be expected. This resulted in a laminar flow in the sampling line, which, in contrast to turbulent flow results in reduced wall losses of particles < 50 nm (Birmili et al. 2007; Nemitz et al. 2008). Another opportunity to reduce wall losses in the measurement setup is using a shorter sampling line (e.g. Hinds, 1999). However, due to the weight of the spectrometer (32 kg) it is not possible to install it atop the 10 m hydraulic mast in a temperature-controlled housing.
   It is correct that particle attenuation in sampling lines might result in an underestimation of fluxes (Suyker and Verma , 1993). The effect of tube attenuation would be indicated in the cospectrum by a signal attenuation at higher frequencies in comparison to the Kaimal cospectrum (Suyker and Verma, 1993). This behaviour, however, is not evident in the measured cospectra at our site (cf. Figure 3). Whereas the use of turbulent flows is discussed for the EC measurement of particle number fluxes (Buzorius et al. 1998), a number of (recent) papers also designed setups using laminar flow conditions (Buzorius et al. 1998; Deventer et al., 2013, 2015, 2018).

4) *L89 – How long is the bypass line, what is the estimated delay time between the inlet and the instrument?*
   The sampling line has a length of 10.75 m from the sample inlet to the particle spectrometer. We refer to the subject of 'time delay' in the comment below (Comment 7).

5) *Figure 1.0 – Is it possible to elaborate on green areas? Are these fields, forests, or parks?*
   Thank you for your question. As already stated in the response to Reviewer #1, we added some information about the green areas to describe the subtypes of green areas more specifically.

6) *L104 – How many particle number size distributions were gap filled and how many were discarded?*
   We added the following sentence to the revised manuscript:

   **L104 (Section 2.3, "Data handling"):**
   "We measured a total of $8.43 \times 10^8$ PNSDs during the entire study period, of which 85.4 % were gap-filled, 14.0 % were rejected and 0.6 % were without gaps and did not need to be gap-filled."

7) *L112 – How well did the covariance maximization agree with the calculated time lag? Was covariance maximized for each flux period? The latter can introduce bias to the data.*
A previous analysis of the median time lag due to the results of covariance maximisation resulted in a time delay of about 9 seconds. The calculated time lag for the measuring system due to the flow rates was around 6 seconds, however, excluding the travel time from the EEPS inlet to the electrometers and without accounting for potential pressure losses within the system. Thus, the calculated time lag is somewhat lower but in a similar range. Since Buzorius et al. (1998) state that the cross-correlated delay time (i.e. covariance maximisation) is more reliable as being independent from the sampling flow rate, we specified a time lag window of 9 seconds ± 4.5 seconds for covariance maximisation, which was applied to for each flux period. Thus, we kept the time lag in a physically plausible range. In case no maximisation was found, a time lag of 9 seconds was used.

8) *L171 – Was the EEPS checked with a calibrated DMA to ensure sizing accuracy?*
Yes, the instrument was factory-calibrated before the start of the measurement campaign. Additionally, we conducted a laboratory comparison to a reference size spectrometer at the European Centre for Aerosol Calibration and Characterisation at TROPOS, Leipzig (Germany). Integrated deviations indicated overestimation of the EEPS concentration by + 26.0% for NUC, + 9.7% for AIT and + 20.8% for ACC particles. Unfortunately, the comparison cannot be directly compared to the measurements conducted in this study due to two reasons:
- The data of the EEPS laboratory comparison was not gap-filled as we compared average size spectra to the TROPOS reference spectrometer (12 hour average of ambient nocturnal aerosol). The EEPS minimum thresholds of the size bins, which are the criteria for gap-filling a concentration in a certain size bin or not (cf. Meyer-Kornblum et al., 2019), declines with increasing averaging interval since the random noises of the electrometers cancel each other out over a long averaging period. Thus, gap-filling was not necessary. However, the impact of gap-filling on the PNSD could not be investigated so that results of the laboratory comparison cannot be directly transferred.
- Generally, it is possible to apply the gap-filling procedure to the 10 Hz EEPS data of the laboratory comparison to include the impact of gap-filling. However, the laboratory comparison measurements were conducted during the night with lower concentrations (2650 cm$^{-3}$, 10 nm < $D_p$ < 200 nm) than at the measurement site in Berlin (~ 8000 cm$^{-3}$). Hence, the gap-filling would probably be related to higher uncertainty than the gap-filling conducted to data from the Berlin site as lower concentrations usually show a higher number of gaps in the 10 Hz data. Hence, comparability to the laboratory EEPS data is not feasible, even after the gap-filling procedure.

9) *L203 – Similar to Figure 1.0, can you better define green area?*
We revised the source-area weighted contribution of land-use to include the different types of green areas. As a result, also Figures 4 and 5 were updated (cf. Figures R4 and R5). We revised the respective paragraphs for clarification:

**L148 (Section 2.6, "Land-use regression analysis"):**
"We binned data into 16 wind sectors, which consisted of three land-use types, namely "built-up areas", "traffic areas", and "green areas". Green areas (cf. Figure 1) were not classified into more specific subtypes since the surface fractions of forest, bushes/single trees, and garden areas in the flux footprints were small (cf. Figure 5). Additionally, the available data on specific subtypes of green areas was limited and could not differentiate, for instance, between grass areas with or without single trees."
**L202 (Section 3.4, "Footprint analysis"):**
"A detailed analysis of source-area weighted contribution of land-use indicates 60 % of built-up areas, 26 % traffic areas, 11 % green areas, and 2 – 3 % water surfaces in the flux footprint (Figure 5). Green areas were mainly characterised by grass with or without single trees whereas forests, bushes/single trees, and gardens contribute little to the flux footprint."

[Figure]

**Figure R4:** Footprint climatology of (a) the entire three-year period, (b) the first year, (c) the second year, and (d) the third year. Flux footprints were calculated for a 4 km x 4 km area (4 m spatial resolution). For reasons of clarity only the 40 %, 50 % (purple), 60 %, and 80 % contour lines are shown (footprint model: Kljun et al.,2015; map data sources: Geoportal Berlin, 2014 (modified), 2021; Umweltatlas Berlin, 2017).

[Figure]

**Figure R5:** Amount of land-use contributing to the flux footprint climatologies separated according to the different time periods of measurement. For data analysis, the footprint information in a 4 km x 4 km grid centred on the measurement site was used (4 m spatial resolution). This footprint area completely includes the 80 % contour line.

10) *L285 to L297 – This section was challenging to follow. The reference to the figure helped, but I think some work could be done to really highlight the key observations that can be made from these wind-sector plots.*

Thank you very much for that comment. We rephrased this section, also considering the comment of Referee #1:

**L292 (Section 3.7, "Wind sector analysis of modal particle number fluxes"):**

"When net fluxes were distinguished into emission and deposition fluxes, the emission fluxes indicated similar behaviour such as the average $F_{TNC}$ due to the high frequency of emission events (cf. chapter 3.5, Figure 12a). Strongest emission fluxes occurred for wind from W to NE whereas the most frequent emission fluxes were evident for wind from 180 to 315° (Figure 12c). Strongest deposition fluxes prevailed under north-westerly wind whereas lower deposition was evident for the remaining directions (Figure 12b). However, the most frequent deposition fluxes were observed under winds from southerly and north-westerly directions (Figure 12d). Since particle deposition occurred most often during night (cf. Figure 9b), the wind direction related frequency distribution of deposition fluxes (Figure 12d) was similar to the night time wind rose (not shown here) which was characterised by most frequent winds from southerly and north-westerly directions. The wind direction related frequency distribution of emission fluxes, however, was similar to the daytime wind rose (not shown here). Generally, ACC particles were more frequently characterised by deposition fluxes than other particle modes."

**References**

Birmili, W.; Stopfkuchen, K.; Hermann, M.; Wiedensohler, A.; Heintzenberg, J.: Particle Penetration Through a 300 m Inlet Pipe for Sampling Atmospheric Aerosols from a Tall Meteorological Tower, Aerosol Sci. Technol., 41 (9), 811–817, doi: 10.1080/02786820701484948, 2007.

Buzorius, G.; Rannik, Ü.; Mäkelä, J. M.; Vesala, T.; Kulmala, M.: Vertical aerosol particle fluxes measured by eddy covariance technique using condensational particle counter, J Aerosol Sci, 29 (1-2), 157–171, doi: 10.1016/S0021-8502(97)00458-8, 1998.

Deventer, M. J.; El-Madany, T.; Griessbaum, F.; Klemm, O.: One-year measurement of size-resolved particle fluxes in an urban area, Tellus B Chem. Phys. Meteorol., 67 (1), 25531, doi: 10.3402/tellusb.v67.25531, 2015.

Deventer, M. J.; Griessbaum, F.; Klemm, O.: Size-resolved flux measurement of sub-micrometer particles over an urban area, metz, 22 (6), 729–737, doi: 10.1127/0941-2948/2013/0441, 2013.

Deventer, M. J.; Heyden, L. von der; Lamprecht, C.; Graus, M.; Karl, T.; Held, A.: Aerosol particles during the Innsbruck Air Quality Study (INNAQS). Fluxes of nucleation to accumulation mode particles in relation to selective urban tracers, Atmospheric Environ., 190, 376–388, doi: 10.1016/j.atmosenv.2018.04.043, 2018.

Geoportal Berlin: Biotoptypen (Umweltatlas), license „dl-de/by-2-0" (www.govdata.de/dl-de/by-2-0), URL: https://fbinter.stadt-berlin.de/fb/berlin/service_intern.jsp?id=s_fb_berlinbtk@senstadt&type=WFS, last access: 20 May 2021, 2014.

Hinds, W. C.: Aerosol technology. Properties, behavior, and measurement of airborne particles, 2. ed., Wiley, New York, 1999.

Kljun, N.; Calanca, P.; Rotach, M. W.; Schmid, H. P.: A simple two-dimensional parameterisation for Flux Footprint Prediction (FFP), Geosci. Model Dev., 8 (11), 3695–3713, doi: 10.5194/gmd-8-3695-2015, 2015.

Meyer-Kornblum, A.; Gerling, L.; Weber, S.: Gap-filling Fast Electrical Mobility Spectrometer Measurements of Particle Number Size Distributions for Eddy Covariance Application, Aerosol Air Qual. Res., 19 (12), 2721–2731, doi: 10.4209/aaqr.2019.06.0291, 2019

Nemitz, E.; Jimenez, J. L.; Huffman, J. A.; Ulbrich, I. M.; Canagaratna, M. R.; Worsnop, D. R.; Guenther, A. B.: An Eddy-Covariance System for the Measurement of Surface/Atmosphere Exchange Fluxes of Submicron Aerosol Chemical Species—First Application Above an Urban Area, Aerosol Sci Technol, 42 (8), 636–657, doi: 10.1080/02786820802227352, 2008.

Suyker, A. E.; Verma, S. B.: Eddy correlation measurement of CO2 flux using a closed-path sensor: Theory and field tests against an open-path sensor, Boundary Layer Meteorol., (64), 391–407, 1993.

Umweltatlas Berlin: Verkehrsmengen DTV 2014 (Umweltatlas), license „dl-de/by-2-0" (www.govdata.de/dl-de/by-2-0), URL: https://fbinter.stadt-berlin.de/fb/berlin/service_intern.jsp?id=wfs_07_01verkmeng2014 @senstadt&type=WFS, last access: 20 May 2021, 2017.